# Microglia facilitate and stabilize the response to general anesthesia via modulating the neuronal network in a brain region-specific manner

**Yang He[1†], Taohui Liu[1†], Quansheng He[1†], Wei Ke[1], Xiaoyu Li[1], Jinjin Du[1,2], Suixin Deng[1], Zhenfeng Shu[1], Jialin Wu[1], Baozhi Yang[1,2], Yuqing Wang[1,2], Ying Mao[1], Yanxia Rao[3], Yousheng Shu[1]\*, Bo Peng[1,4]\***

[1]Department of Neurosurgery, Huashan Hospital, Institute for Translational Brain Research, State Key Laboratory of Medical Neurobiology, MOE Frontiers Center for Brain Science, MOE Innovative Center for New Drug Development of Immune Inflammatory Diseases, Fudan University, Shanghai, China; [2]School of Basic Medical Sciences, Jinzhou Medical University, Jinzhou, China; [3]Department of Neurology, Zhongshan Hospital, Department of Laboratory Animal Science, MOE Frontiers Center for Brain Science, Fudan University, Shanghai, China; [4]Co-Innovation Center of Neurodegeneration, Nantong University, Nantong, China

**\*For correspondence:**
yousheng@fudan.edu.cn (YS);
peng@fudan.edu.cn (BP)

[†]These authors contributed equally to this work

**Competing interest:** The authors declare that no competing interests exist.

**Abstract** General anesthesia leads to a loss of consciousness and an unrousable state in patients. Although general anesthetics are widely used in clinical practice, their underlying mechanisms remain elusive. The potential involvement of nonneuronal cells is unknown. Microglia are important immune cells in the central nervous system (CNS) that play critical roles in CNS function and dysfunction. We unintentionally observed delayed anesthesia induction and early anesthesia emergence in microglia-depleted mice. We found that microglial depletion differentially regulates neuronal activities by suppressing the neuronal network of anesthesia-activated brain regions and activating emergence-activated brain regions. Thus, microglia facilitate and stabilize the anesthesia status. This influence is not mediated by dendritic spine plasticity. Instead, it relies on the activation of microglial P2Y12 and subsequent calcium influx, which facilitates the general anesthesia response. Together, we elucidate the regulatory role of microglia in general anesthesia, extending our knowledge of how nonneuronal cells modulate neuronal activities.

## eLife assessment

This study presents a **valuable** finding on the mechanisms underlying general anesthesia, with a focus on microglial regulation. The evidence supporting the claims of the authors is **solid**, although some of the novelty of these findings may be reduced based on the recent publication of a similar study. The work will be of interest to medical biologists working on mechanisms of anesthesia, microglia, and neuron–microglia interaction.

## Introduction

General anesthesia is a cornerstone of modern medical sciences. Upon the use of anesthetics, patients lose consciousness and enter an unarousable state. Although general anesthetics are widely used in clinical procedures, the mechanism of general anesthesia remains elusive. Different anesthetics

activate or inhibit specific receptors in neurons, modulating neuronal activities across the entire network. However, the involvement of nonelectrically active glial cells in anesthesia is poorly understood. Microglia are yolk sac-derived glial cells in the central nervous system (CNS) (*Ginhoux et al., 2010*). They play critical roles in CNS development, function, and dysfunction (*Brioschi et al., 2020*; *Prinz et al., 2019*). Previous studies have found that neuronal activity in the CNS network modulates microglial activity. Microglia exhibit elevated process motility, extension, and territory surveillance during anesthetization and sleep (*Liu et al., 2019a*; *Stowell et al., 2019*). Suppression of neuronal activity increases calcium signaling in microglial processes (*Umpierre et al., 2020*). Conversely, microglia also modulate neuronal activity via multiple mechanisms (*Sipe et al., 2016*; *Stevens et al., 2007*; *Favuzzi et al., 2021*; *Schafer et al., 2012*; *Eyo et al., 2017*; *Badimon et al., 2020*). The microglial regulation of neuronal activities thus raises the question of whether microglia can modulate the general anesthesia response.

Microglial survival relies on colony-stimulating factor 1 receptor (CSF1R) signaling (*Erblich et al., 2011*). Pharmacological inhibition of CSF1R efficiently eliminates CNS microglia (*Elmore et al., 2015*; *Huang et al., 2018b*; *Zhou et al., 2022*; *Rao et al., 2021*; *Xu et al., 2020*; *Huang et al., 2018a*). Previous studies have shown that acute microglial depletion does not induce neuroinflammation (*Elmore et al., 2015*; *Huang et al., 2018b*; *Huang et al., 2018a*; *Elmore et al., 2014*). The ablation of microglia in adulthood also does not result in obvious general behavioral dysfunctions (*Elmore et al., 2015*; *Elmore et al., 2014*), although this finding is controversial (*Parkhurst et al., 2013*). It seems that microglia are disposable under physiological conditions. However, when we killed microglia with PLX5622, a CSF1R inhibitor (*Spangenberg et al., 2019*), we unintentionally observed robust resistance to anesthetic administration. This suggests that microglia may facilitate general anesthesia by modulating neuronal network activity.

To this end, we first quantified the influence of microglial depletion on the response to general anesthesia. We utilized the loss of righting reflex (LORR) and recovery of righting reflex (RORR) to evaluate anesthesia induction and emergence, respectively. After microglial depletion, mice displayed a longer LORR time and a shorter RORR time. The dampened general anesthesia response was not dependent on specific anesthetics or receptors as this phenomenon was observed with three different agonists of the GAGA$_A$ receptor (pentobarbital, propofol, and chloral hydrate) and one antagonist of the NMDA receptor (ketamine). Electroencephalography (EEG) and electromyography (EMG) findings further confirmed our initial observation. Different brain regions diversely regulate anesthesia induction and emergence. Anesthesia induction is positively correlated with anesthesia-activated brain regions (AABRs). In contrast, anesthesia emergence is positively correlated with emergence-activated brain regions (EABRs). We observed that microglia modulate brain network activity in a brain region-specific manner rather than in a universal manner for all brain regions. Based on c-Fos reactivity and patch-clamp recordings, we demonstrated that microglial depletion inhibits AABRs and activates EABRs. The divergent effects in different brain regions orchestrate the status during general anesthesia use. Microglia-mediated anesthesia modulation is not attributed to dendritic spine plasticity. We found that mice with genetic knockout or pharmacological inhibition of microglial P2Y12 were more resistant to general anesthesia. In addition, the contribution of microglial P2Y12 to anesthesia response was further confirmed by the mouse receiving microglia replacement, in which the replaced microglia-like cells are P2Y12$^-$ (*Xu et al., 2020*). On the other hand, the intracellular Ca$^{2+}$ concentration in microglia facilitates and stabilizes the response to general anesthesia. Because purinergic activation of P2Y12 increases intracellular Ca$^{2+}$ (*Jiang et al., 2017*; *Jairaman et al., 2022*; *Pozner et al., 2015*), our results reveal that the general anesthesia response is regulated through P2Y12 to Ca$^{2+}$ signaling in microglia.

In conclusion, our study demonstrates a regulatory role of microglia in the response to general anesthesia and identifies the underlying mechanism of this process. This study extends our knowledge of how nonelectrically active glial cells regulate the general anesthesia response. It also sheds new light on how microglia contribute to maintaining the status of the brain network. When we were preparing our manuscript, a paper discussing a similar topic emerged (*Cao et al., 2023*).

**A** PLX5622 depletion | CD repopulation

8-week-old | D0 | D14 | D35

**B**

**Pentobarbital**

LORR (s): P = 0.5455, P = 0.0434, P = 0.0380; D0 D14 D35

RORR (min): P = 0.9912, P = 0.0044, P = 0.0008; D0 D14 D35

**Propofol**

LORR (s): P = 0.0022, P = 0.1028, P = 0.0257; D0 D14 D35

RORR (min): P = 0.1916, P = 0.0004, P = 0.0190; D0 D14 D35

**Chloral hydrate**

LORR (s): P = 0.8986, P = 0.5029, P = 0.0002; D0 D14 D35

RORR (min): P = 0.7139, P = 0.3279, P = 0.3024; D0 D14 D35

**Ketamine**

LORR (s): P = 0.1249, P = 0.4200, P = 0.0571; D0 D14 D35

RORR (min): P = 0.0378, P = 0.0026, P = 0.0074; D0 D14 D35

**Figure 1.** Microglial depletion impedes anesthesia induction and accelerates emergence. (**A**) Scheme of time points for microglial depletion and repopulation by PLX5622 and CD. (**B**) Mice exhibit delayed induction and early emergence in pentobarbital-, propofol-, chloral hydrate-, and ketamine-induced anesthesia. N = 11, 10, 10, and 12 mice for pentobarbital, propofol, chloral hydrate, and ketamine, respectively. Repeated measures (paired) one-way ANOVA with Geisser–Greenhouse correction and Tukey's multiple-comparison test. Data are presented as mean ± SD. PLX5622: PLX5622-formulated diet; CD: control diet; LORR: loss of righting reflex; RORR: recovery of righting reflex. All animals are male mice.

The online version of this article includes the following source data for figure 1:

**Source data 1.** Raw data for LORR and RORR times.

## Results

### Microglia regulate the induction of and emergence from general anesthesia

We unintentionally observed that mice become more resistant to anesthesia after microglia and macrophage depletion by the CSF1R inhibitor PLX5622. To quantitatively study whether microglial depletion indeed influences the induction of and emergence from general anesthesia, we first fed mice a PLX5622-formulated diet (PLX5622 hereafter) to ablate CNS microglia and peripheral macrophages (*Zhou et al., 2022*; *Xu et al., 2020*; *Huang et al., 2018a*; *Yang et al., 2020*). After 14 d of PLX5622 administration, we intraperitone-ally injected pentobarbital (80 mg pentobarbital

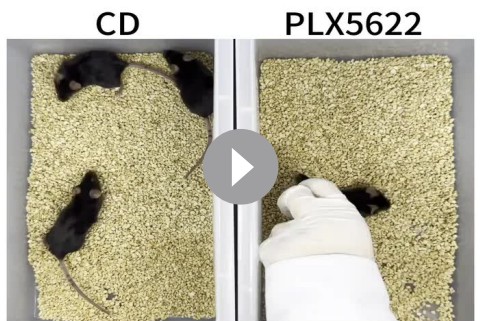

**Video 1.** PLX5622-treated mice displayed a longer time for LORR compared to the naïve mice.
https://elifesciences.org/articles/92252/figures#video1

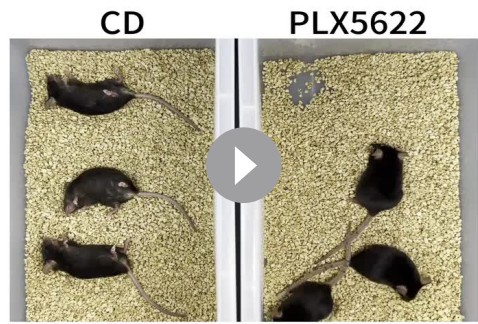

**Video 2.** PLX5622-treated mice displayed a shorter time for RORR compared to the naïve mice.
https://elifesciences.org/articles/92252/figures#video2

sodium per kg of body weight) into the PLX5622-treated mice (*Figure 1A*). Compared to naïve mice on day 0 (D0), the PLX5622-treated mice at D14 displayed a longer time for LORR and shorter time for RORR (*Figure 1B* and *Videos 1 and 2*). Pentobarbital is an agonist of the GABA$_A$ receptor (*Sykes and Thomson, 1989*). We next examined whether this microglia- and macrophage-mediated regulation of the anesthesia response is restricted to pentobarbital or GABA$_A$ receptor agonists. We assessed LORR and RORR in PLX5622-treated mice by using other anesthetics, including two other GABA$_A$ receptor agonists (propofol, 200 mg/kg of body weight; chloral hydrate, 400 mg/kg of body weight) (*Sebel and Lowdon, 1989*; *Lovinger et al., 1993*; *Garrett and Gan, 1998*) and one NMDA receptor antagonist (ketamine, 100 mg/kg of body weight) (*Anis et al., 1983*). Similar trends were observed in propofol-, chloral hydrate-, and ketamine-induced LORR and RORR (*Figure 1B*). To exclude the possibility that PLX5622-induced anesthesia resistance results from tolerance to repetitive anesthetic injection, we sequentially treated mice with the same anesthetics five times at 7-day intervals (*Figure 2A*). Both LORR and RORR were unchanged for pentobarbital, propofol, chloral hydrate, and ketamine, except for chloral hydrate-induced LORR after D14 (the third dose) (*Figure 2B*). PLX5622-treated mice exhibited a resistant phenotype (*Figure 1B*). In contrast, the third dose of chloral hydrate made mice more susceptible to anesthesia (*Figure 2B*), an

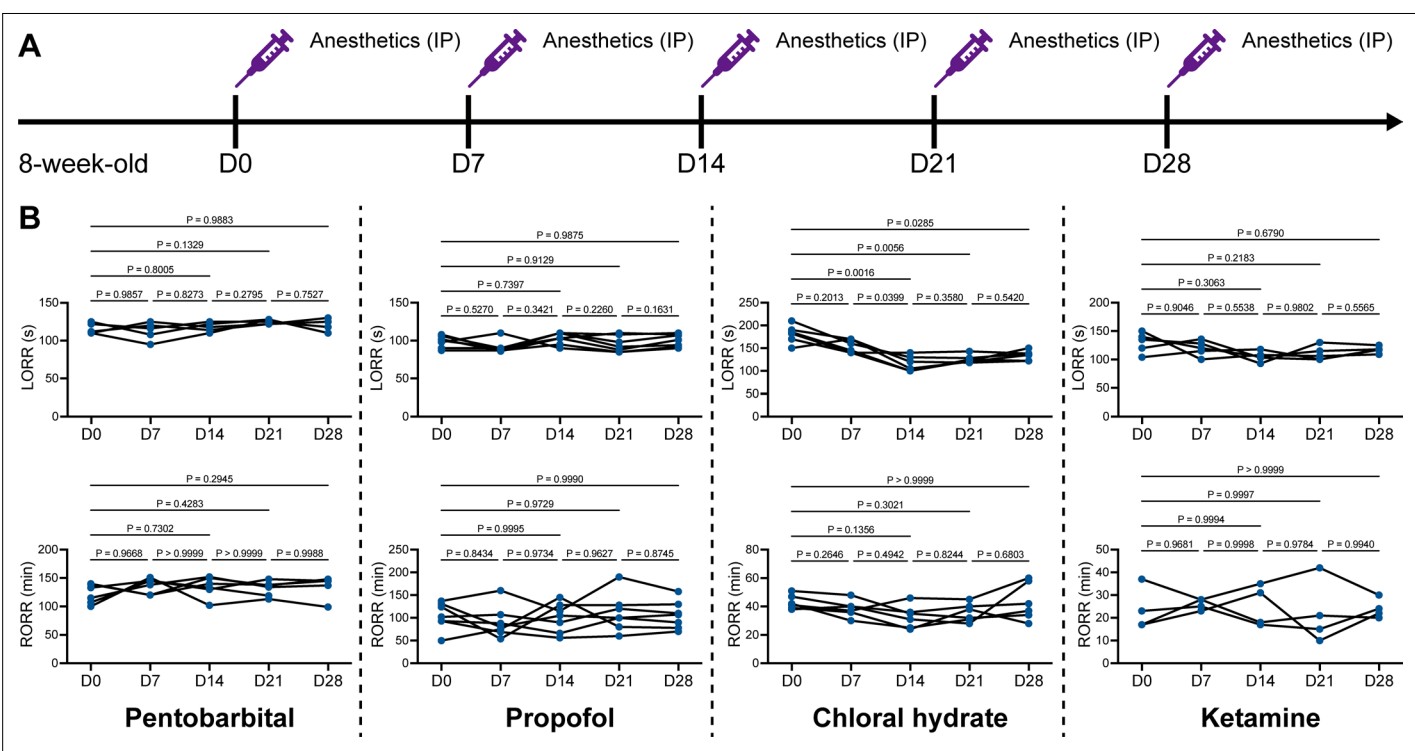

**Figure 2.** Repetitive anesthetic treatment does not result in anesthesia tolerance. (**A**) Scheme of time points for anesthetic treatments and righting reflex examination. (**B**) Repetitive treatment with pentobarbital, propofol, chloral hydrate or ketamine does not induce anesthesia tolerance in mice. N = 6, 7, 6, and 5mice are treated with pentobarbital, propofol, chloral hydrate, and ketamine, respectively. Repeated measures (paired) one-way ANOVA with Geisser–Greenhouse correction and Tukey's multiple-comparison test. LORR: loss of righting reflex; RORR: recovery of righting reflex. All animals are male mice.

The online version of this article includes the following source data for figure 2:

**Source data 1.** Raw data for LORR and RORR times.

opposite trend from PLX5622 treatment. Consequently, PLX5622-induced anesthesia resistance is not attributed to the tolerance of repetitive anesthetic administration.

PLX5622 can simultaneously ablate brain microglia and peripheral macrophages (*Yang et al., 2020*). To exclude the possibility that CSF1R-mediated general anesthesia regulation resulted from peripheral macrophages, we utilized the blood–brain barrier-impermeable CSF1R inhibitor PLX73086 to ablate peripheral macrophages without influencing brain microglia (*Bellver-Landete et al., 2019*). After administration of the PLX73086-formulated diet (PLX73086 hereafter) for 14 d, macrophages in the liver, lung, spleen, and kidney were significantly ablated, while brain microglia were not influenced (*Figure 3A and B*). PLX73086-treated mice (peripheral macrophage-depleted, brain microglia-unchanged) displayed similar general anesthetic responses as naïve mice at D0, including those treated with pentobarbital, propofol, chloral hydrate, and ketamine, except for the RORR in pentobarbital-treated mice (*Figure 3C*). Since PLX73086-treated mice (peripheral macrophage-depleted, brain microglia-unchanged) exhibited a delayed RORR with pentobarbital administration (*Figure 3C*), whereas PLX5622-treated mice (peripheral macrophage-depleted, brain microglia-depleted) displayed an earlier RORR (*Figure 1B*), the pentobarbital-induced early emergence in PLX5622-administered mice is not attributed to peripheral macrophage ablation. Therefore, microglial depletion, rather than macrophage depletion, leads to resistance to general anesthesia.

Next, we reasoned whether this anesthesia resistance is permanent or can be reversed by microglial repopulation. To address this question, we ceased CSF1R inhibition by treating the microglia-depleted mice with a control diet (CD) for 21 d to allow microglia to repopulate the brain (*Figure 1A*), at which point repopulated microglia recovered to the same density and similar transcriptional characteristics as those in control mice (*Huang et al., 2018a*). The LORR and RORR of microglia-repopulated mice at D35 recovered to the same level as those of naïve mice at D0 (*Figure 1B*), indicating that fully repopulated microglia can reverse the anesthesia susceptibility of mice.

Together, our results indicate that microglial depletion by inhibiting CSF1R results in delayed anesthesia induction and early anesthesia emergence, making the animals more resistant to general anesthesia.

To further characterize the impact of microglial depletion throughout the anesthetization window, we recorded EEG and EMG signals to monitor the anesthesia state before and after pentobarbital administration (*Figure 4A*). Microglial depletion showed no obvious influence on the EEG in the awake or conscious state before pentobarbital administration (*Figure 4B*). In contrast, microglia-depleted mice exhibited delayed anesthesia induction and early emergence in response to pentobarbital (*Figure 4B and C*). In addition, microglial depletion significantly altered the power spectrum during anesthesia induction and emergence but not consciousness (*Figure 4D*). The EMG results showed that muscular activity in the conscious state was unchanged upon microglial depletion. In contrast, microglia-depleted mice exhibited a delayed loss and early recovery of muscular activity after pentobarbital injection (*Figure 4E*). Moreover, the probability of being in the conscious state, predicted by an algorithm combining EEG and EMG (*Zhai et al., 2023*), triple confirmed delayed anesthesia induction and early emergence after microglial depletion (*Figure 4F*). Similar results were also observed for propofol (*Figure 5*) and ketamine (*Figure 6*). The EEG and EMG results demonstrate that microglial depletion impedes the anesthesia process.

Together, our results demonstrate that brain microglia-depleted mice are resistant to general anesthetics. In other words, microglia play important roles in facilitating and stabilizing the status of general anesthesia response.

## Microglia facilitate the anesthesia response in a brain region-specific manner

A previous study indicated that microglia negatively regulate neuronal activity through the microglial catabolism of ATP and neuronal adenosine receptor $A_1R$. Microglial depletion enhances neuronal activity in the striatum (*Badimon et al., 2020*). Different brain regions regulate anesthesia induction and emergence in diverse manners. If microglial depletion indiscriminately influences neuronal activities among different brain regions, the enhanced activities in AABRs and EABRs would be mutually antagonistic, complicating the anesthetic effect. To investigate whether microglia regulate neuronal activity in an indiscriminate or brain region-specific manner, we examined c-Fos expression in AABRs and EABRs of CD- and PLX5622-treated mice (*Figure 7A*). We first studied AABRs, including the lateral

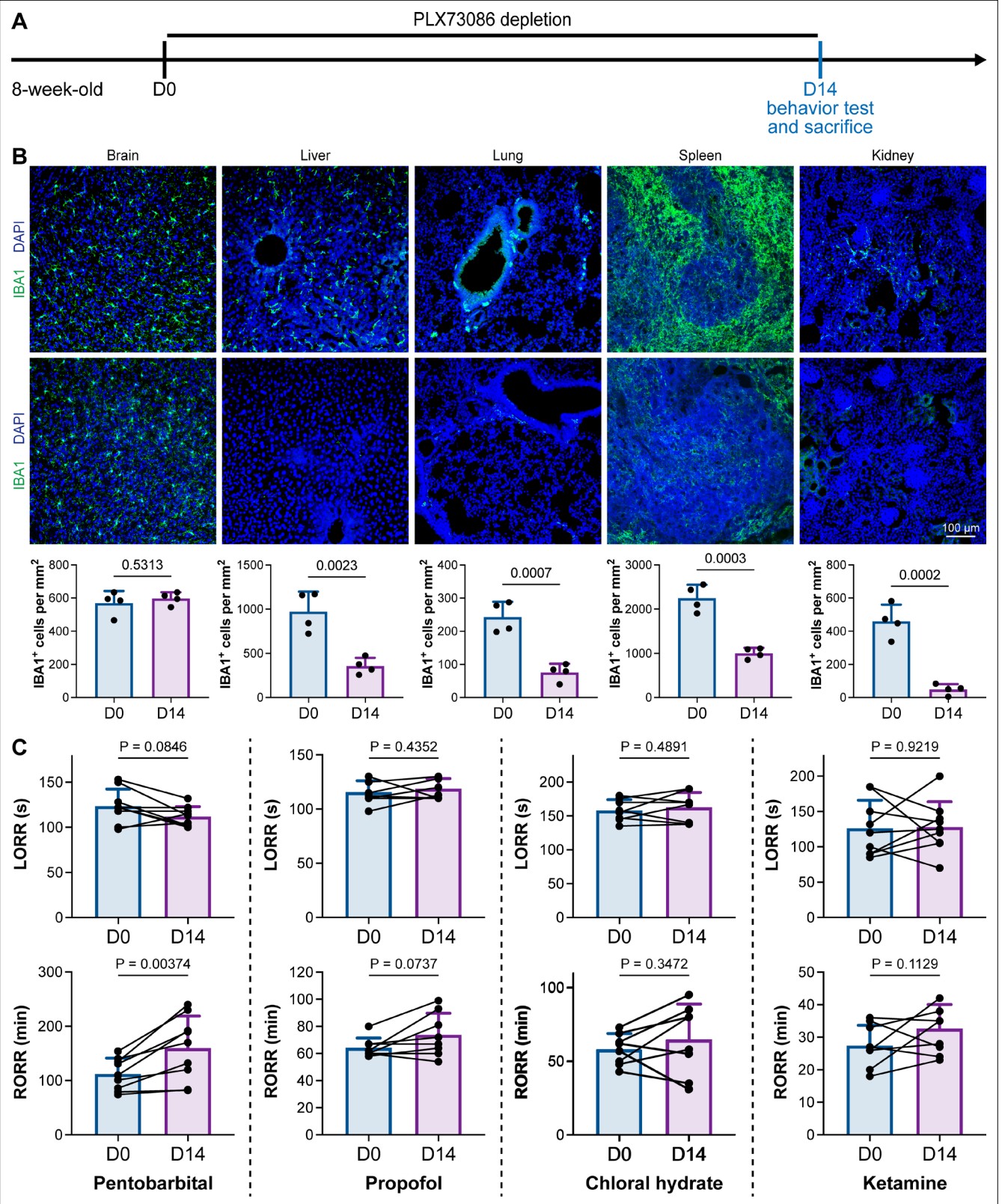

**Figure 3.** CSF1R inhibition-induced general anesthesia regulation is not due to the depletion of peripheral macrophages. (**A**) Scheme of time points for peripheral macrophage depletion by PLX73086. (**B**) CSF1R inhibition by PLX73086 dramatically ablates macrophages in the liver, lung, spleen, and kidney and does not ablate brain microglia. N = 4mice for each group. Two-tailed independent t-test. (**C**) Depletion of peripheral macrophages does not influence the anesthesia induction of pentobarbital, propofol, chloral hydrate, and ketamine or the emergence from propofol, chloral hydrate,

*Figure 3 continued on next page*

Figure 3 continued

and ketamine. However, it impedes anesthesia emergence from pentobarbital. N = 9, 8, 8, and 9mice for pentobarbital, propofol, chloral hydrate, and ketamine, respectively. Two-tailed paired *t*-test. Data are presented as mean ± SD. PLX73086: PLX73086-formulated diet; LORR: loss of righting reflex; RORR: recovery of righting reflex. All animals are male mice.

The online version of this article includes the following source data for figure 3:

Source data 1. Assessment of PLX73086 influence on IBA1⁺ cell density in different organs.

Source data 2. Raw data for LORR and RORR times.

habenula (LHb) (*Gelegen et al., 2018*; *Zhou et al., 2023*), supraoptic nucleus (SON) (*Jiang-Xie et al., 2019*), ventrolateral preoptic nucleus (VLPO) (*Nelson et al., 2002*), and thalamic reticular nucleus (TRN) (*Zhang et al., 2019a*). The abundance of c-Fos⁺ cells was significantly reduced in the LHb and SON of microglia-depleted mice (*Figure 7B*). VLPO exhibited a decreasing trend in the abundance of c-Fos⁺ cells, although it did not reach a statistically significant level (p=0.1592) (*Figure 7B*). The abundance of c-Fos⁺ cells was unchanged in the TRN (*Figure 7B*). We next examined EABRs, including the paraventricular thalamus (PVT) (*Wang et al., 2023*), locus coeruleus (LC) (*Vazey and Aston-Jones, 2014*), lateral hypothalamus (LH) (*Venner et al., 2016*; *Zhao et al., 2021*), and ventral tegmental area (VTA) (*Solt et al., 2014*; *Taylor et al., 2013*). In contrast to a suppressed trend in AABRs, neuronal activity exhibited an enhanced trend in EABRs. Microglial depletion significantly increased the c-Fos⁺ cell number in the PVT and LC (*Figure 7C*). c-Fos⁺ cell numbers were also increased in the LH and VTA, although the difference did not reach statistical significance (p=0.0598 and 0.1436, respectively) (*Figure 7C*). To exclude the possibility that the different c-Fos⁺ cell numbers were attributed to animal handling, we compared c-Fos expression in saline-injected and noninjected mice (*Figure 8A*). Our results indicate that animal handling (saline injection) did not influence c-Fos expression in the LHb, SON, VLPO, TRN, PVT, LC, LH, or VTA (*Figure 8B and C*). Therefore, we found that microglial depletion negatively regulates AABRs and positively regulates EABRs, indicating that microglia regulate neuronal activity in a brain region-specific manner. The suppressed neuronal activity in AABRs leads to delayed anesthesia induction. In contrast, the elevated neuronal activity in EABRs results in the early emergence of anesthesia.

The protein expression of c-Fos is relatively slow, peaking at a timepoint hours after transcription (*Xiu et al., 2014*). Mice in our study were quickly sacrificed after deep anesthesia, typically within 5–10 min. The abundance of c-Fos protein seen with immunostaining reflected neuronal activity during the consciousness stage (*Figure 7*). In contrast, the mRNA expression of *Fos* (encoding c-Fos) is relatively fast, peaking at approximately 30 min after induction (*Xiu et al., 2014*). We asked how microglia influence neuronal activity during the anesthesia stage and whether microglia differentially influence neuronal activity between the consciousness and anesthesia stages. To this end, we sacrificed microglia-naïve and microglia-depleted mice 30 min after deep anesthetization by pentobarbital and simultaneously labeled the c-Fos protein by immunostaining and *Fos* mRNA by RNAscope (*Figure 9A*). The c-Fos⁺ cells represent activated neurons during the consciousness stage, while *Fos*⁺ cells represent activated neurons during the anesthesia stage (*Figure 9A*). We compared the c-Fos⁺ and *Fos*⁺ cells in AABRs and EABRs in which neuronal activity was significantly altered in microglia-depleted mice, including the LHb, SON, PVT, and LC (*Figures 7B and C and 9B and C*). After microglial depletion, activated neurons in the anesthesia stage (*Fos*⁺) displayed similar trends as those in the consciousness state (c-Fos⁺) in the LHb, SON, and LC (*Figure 9B and C*). However, the number of anesthesia-activated neurons (*Fos*⁺) was unchanged between the naïve and microglia-depleted PVT, whereas the number of consciousness-activated neurons was significantly increased upon microglial depletion (*Figure 9C*). Exploiting the c-Fos protein and *Fos* mRNA dual labeling, we further compared consciousness-activated anesthesia-activated (c-Fos⁺ *Fos*⁺), consciousness-activated anesthesia-nonactivated (c-Fos⁺ *Fos*⁻), and consciousness-nonactivated anesthesia-activated (c-Fos⁻ *Fos*⁺) neurons between naïve and microglia-depleted mice (*Figure 9A*). In the LHb and SON of AABRs, consciousness-activated anesthesia-activated (c-Fos⁺ *Fos*⁺), consciousness-activated anesthesia-nonactivated (c-Fos⁺ *Fos*⁻), and consciousness-nonactivated anesthesia-activated (c-Fos⁻ *Fos*⁺) cell numbers exhibited a decreasing trend after microglial depletion (*Figure 9B*). This indicates that microglial depletion influences AABR neuronal activity at both the consciousness and anesthesia stages. In the PVT of the EABR, consciousness-activated anesthesia-nonactivated (c-Fos⁺ *Fos*⁻) cell

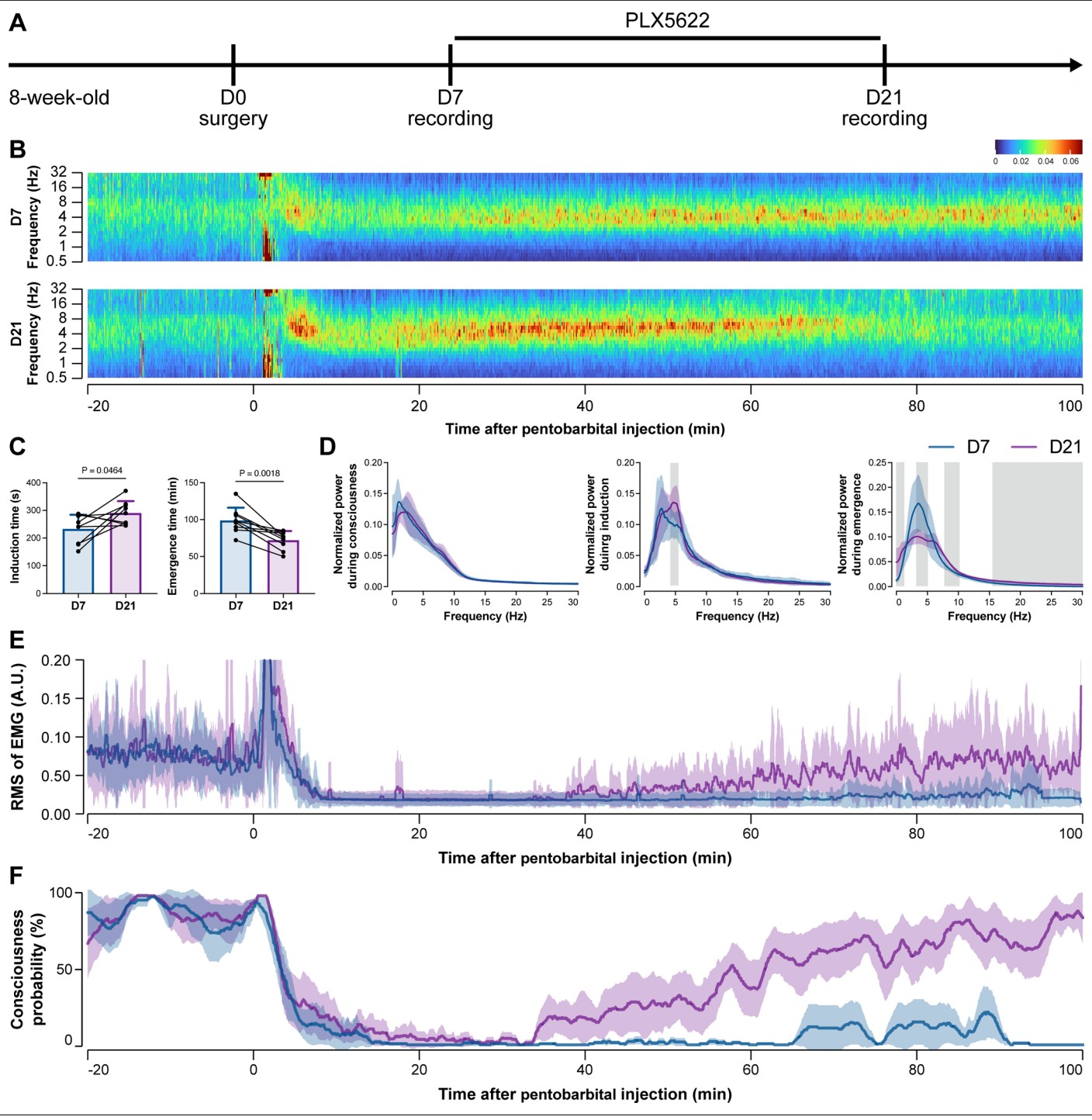

**Figure 4.** Electroencephalography (EEG) and electromyography (EMG) recordings reveal that mice with microglial depletion are resistant to general anesthesia by pentobarbital. (**A**) Scheme of time points for animal surgery, microglial depletion, and EEG/EMG recording. (**B–D**) Microglial depletion shows no obvious change in EEG before the injection of pentobarbital. Instead, it influences the EEG in anesthesia induction and emergence. Two-tailed paired *t*-test. The gray area in (**D**) indicates p<0.05 between D7 and D21. (**E**) Microglial depletion does not change the EMG before the injection of pentobarbital. Instead, it influences the EMG in the anesthesia process. (**F**) Microglial depletion does not change the probability of consciousness before the injection of pentobarbital. Instead, it influences the consciousness probability in the anesthesia process. N = 9mice for each group. Data are presented as mean ± SD. RMS: root mean square; A.U.: arbitrary unit; PLX5622: PLX5622-formulated diet. All animals are male mice.

The online version of this article includes the following source data for figure 4:

**Source data 1.** Raw data for induction and emergence times.

*Figure 4 continued on next page*

*Figure 4 continued*

**Source data 2.** Raw data for the normalized power during different stages.

**Source data 3.** Raw data for the RMS of EMG.

**Source data 4.** Raw data for the consciousness probability.

numbers were significantly increased in microglia-depleted brains, whereas consciousness-activated anesthesia-activated (c-Fos$^+$ Fos$^+$) and consciousness-nonactivated anesthesia-activated (c-Fos$^-$ Fos$^+$) cell numbers were unchanged (*Figure 9C*). This indicates that microglial depletion influences PVT neuronal activity at the consciousness stage but not at the anesthesia stage. In the LC of the EABR, microglial depletion significantly increased the cell numbers of consciousness-activated anesthesia-activated (c-Fos$^+$ Fos$^+$) and consciousness-activated anesthesia-nonactivated (c-Fos$^+$ Fos$^-$) neurons (*Figure 9C*). In contrast, the number of consciousness-nonactivated anesthesia-activated (c-Fos$^-$ Fos$^+$) cells was not altered (*Figure 9C*). This finding indicates that microglial depletion does not influence the LC neurons that are not activated in the consciousness stages.

The results indicate that microglia diversely regulate neuronal activity through a sophisticated brain region-specific manner instead of via indiscriminately negative feedback control as in the striatum (*Badimon et al., 2020*). This may be due to microglial heterogeneity, different neuronal cell types, and/or circuitries in different brain regions.

## Microglial depletion reduces the E/I ratio in AABR but enhances the E/I ratio in EABR

To understand how microglia reshape neuronal activity, we treated mice with CD or PLX5622 for 14 d and performed whole-cell recordings in neurons of SON and LC in acute brain slices, representing AABR and EABR with reduced and increased neuronal activity upon microglial depletion, respectively (*Figure 10A*). We delivered electrical pulses (0.1 ms in pulse duration, 20 Hz, eight pulses) with a current intensity increment of 10 µA every 10 s to the neighboring tissue (approximately 50 µm from the recorded cell) to induce postsynaptic responses, including both evoked excitatory postsynaptic currents (eEPSCs) and evoked inhibitory postsynaptic currents (eIPSCs). In the SON of AABR, higher stimulation currents induced larger amplitudes of both eEPSCs and eIPSCs (*Figure 10B and C*). The peak amplitudes of eEPSCs in microglia-depleted mice were significantly smaller than those in naïve mice, while the peak amplitudes of eIPSCs showed no significant difference (*Figure 10B and C*). As shown in *Figure 10C*, the excitation received by SON neurons dominated in naïve mice. The E/I ratio was also significantly decreased after microglial depletion, indicating decreased neuronal excitability in the SON (*Figure 10D*). SON neurons with microglia depletion exhibited a significantly increased paired-pulse ratio (PPR) of eEPSCs, while the eIPSC PPR was similar between naïve and microglia-depleted mice (*Figure 10E and F*), indicating a reduction in presynaptic release probability in excitatory synapses. Microglial depletion thus results in a more inhibitory state in the AABR SON. In the LC of EABR, the eEPSC amplitudes induced by higher stimulation currents in microglia-depleted mice were substantially greater than those in naïve mice (*Figure 10G and H*). In contrast, the eIPSC amplitudes showed no significant difference (*Figure 10G and H*). In contrast to that in the SON, the E/I ratio in the LC was significantly enhanced in PLX5622-treated mice (*Figure 10I*), indicating an increase in the excitation of LC neurons. Microglial depletion did not change the eEPSC PPR or eIPSC PPR in the LC (*Figure 10J and K*), representing an unchanged presynaptic release probability in both excitatory and inhibitory synapses. Microglial depletion thus results in a more excitatory state in the EABR LC.

In conclusion, our results reveal that microglial depletion decreases AABR and enhances EABR network activities, explaining delayed anesthesia induction and early emergence.

## Microglia-mediated anesthesia modulation is not attributed to the influence of dendritic spines

Microglia play important roles in spine pruning (*Lui et al., 2016*; *Paolicelli et al., 2011*). We asked whether microglia-mediated anesthesia regulation occurs through the alteration of dendritic spines. We first quantified spine density after microglial depletion for 14 d (*Figure 11A*). Spine density in both apical and basal dendrites of layer V pyramidal cells in the medial prefrontal cortex (mPFC) was not changed in the relatively short period of 14 d (*Figure 11B*), when anesthesia induction and

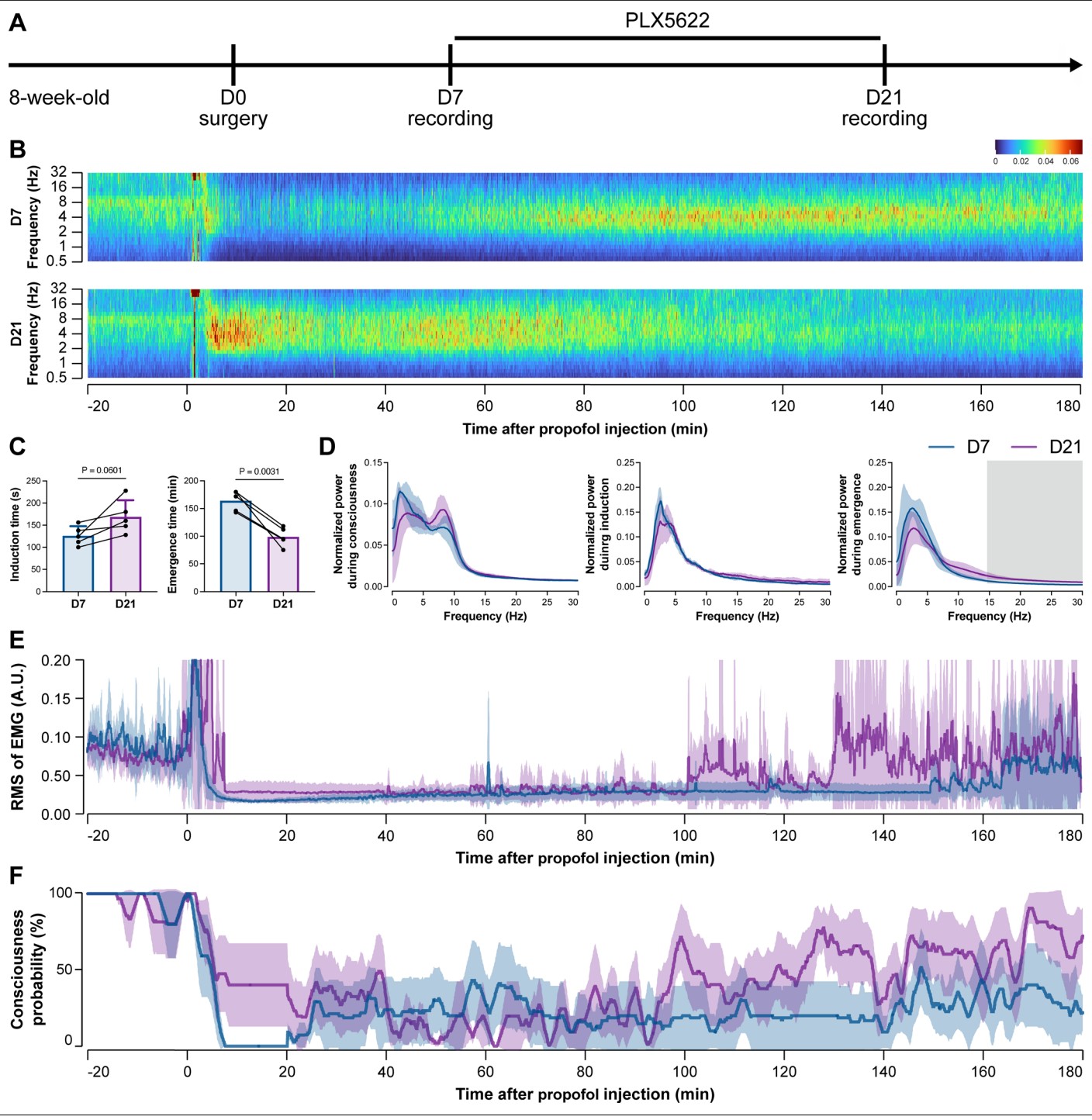

**Figure 5.** Electroencephalography (EEG) and electromyography (EMG) recordings reveal that mice with microglial depletion are resistant to general anesthesia by propofol. (**A**) Scheme of time points for animal surgery, microglial depletion, and EEG/EMG recording. (**B–D**) Microglial depletion does not change the EEG before the injection of propofol. Instead, it influences the EEG in anesthesia induction and emergence. Two-tailed paired *t*-test. The gray area in (**D**) indicates p<0.05 between D7 and D21. (**E**) Microglial depletion does not change the EMG before the injection of propofol. Instead, it influences the EMG in the anesthesia process. (**F**) Microglial depletion does not change the probability of consciousness before the injection of propofol. Instead, it influences the consciousness probability in the anesthesia process. N = 5mice for each group. Data are presented as mean ± SD. RMS: root mean square; A.U.: arbitrary unit; PLX5622: PLX5622-formulated diet. All animals are male mice.

The online version of this article includes the following source data for figure 5:

**Source data 1.** Raw data for induction and emergence times.

*Figure 5 continued on next page*

*Figure 5 continued*

**Source data 2.** Raw data for the normalized power during different stages.

**Source data 3.** Raw data for the RMS of EMG.

**Source data 4.** Raw data for the consciousness probability.

emergence were already robustly influenced (*Figure 1*). The ratios of spine categories of different shapes were altered in microglia-depleted mice. In apical dendrites, the percentage of mature mushroom spines was increased, whereas the percentage of filopodia spines was reduced (*Figure 11B*). In basal dendrites, mature mushroom spines were increased, while thin and filopodia spines were decreased (*Figure 11B*). Thus, short-term microglial depletion for 14 d does not alter spine density but changes the percentage of different categories of spines, even though general anesthesia is dramatically influenced. Next, we explored whether the altered response to general anesthesia resulted from microglia-mediated spine pruning. Microglia phagocytose dendritic spines via the C1q 'eat me' signal (*Stevens et al., 2007*; *Lui et al., 2016*; *Rivest, 2018*). The dendritic spine is remodeled in C1q-dificient mice with increased density and altered spine categories (*Ma et al., 2013*).We thus examined anesthesia induction and emergence in *C1qa*$^{-/-}$ mice (*Figure 11C*), in which the C1q protein complex is formed. Even with the interfered spine pruning, the pentobarbital-, propofol-, or ketamine-induced LORR and RORR were not influenced in C1q-deficient mice (*Figure 11D*). Therefore, our results reveal that even though microglia contribute to spine plasticity, microglia-mediated anesthesia modulation does not result from spine pruning.

## Intracellular calcium in microglia regulates the anesthesia response through P2Y12 signaling

Microglial P2Y12 is a G protein-coupled receptor (GPCR) that modulates neuronal activity (*Eyo et al., 2017*). We thus asked whether microglia-mediated anesthesia modulation is dependent on P2Y12. To address this question, we utilized the selective P2Y12 antagonist 2-MeSAMP (*Srinivasan et al., 2009*) to block P2Y12 signaling by intracranial guide tube implantation (*Figure 12A*). Then, 90 min after 2-MeSAMP administration, brain microglia exhibited a more reactive morphology (*Figure 12B*), the consequence of decreased P2Y12 signaling, as shown in previous studies (*Mastorakos et al., 2021*). Pharmacological inhibition of P2Y12 by 2-MeSAMP delayed the LORR and accelerated the RORR of pentobarbital-induced anesthetization (*Figure 12C*). To further confirm the function of microglial P2Y12 in the anesthesia response, we conditionally knocked out P2Y12 in microglia in *Cx3cr1*$^{+/CreER}$::*P2ry12*$^{fl/fl}$ mice (*Figure 12D*). After four doses of tamoxifen, the majority of P2Y12 was successfully knocked out in *Cx3cr1*$^{+/CreER}$::*P2ry12*$^{fl/fl}$ mice (*Figure 12E*). *Cx3cr1*$^{+/CreER}$::*P2ry12*$^{fl/fl}$ mice exhibited delayed LORR and early RORR in response to pentobarbital (*Figure 12F*), echoing the pharmacological inhibition by 2-MeSAMP. Interestingly, when the conditional knockout of microglial P2Y12 was induced at a lower efficacy in *Tmem119*$^{CreER/CreER}$::*P2ry12*$^{fl/fl}$ mice (*Figure 12G and H*), as the recombinase activity of *Tmem119*-CreER is lower than that of *Cx3cr1*-CreER (*Bedolla et al., 2023*; *Faust et al., 2023*), *Tmem119*$^{CreER/CreER}$::*P2ry12*$^{fl/fl}$ mice displayed an early RORR in response to pentobarbital, but the LORR was not affected (*Figure 12I*). The results indicate that microglia regulate the anesthesia response through P2Y12 signaling in a dose-dependent manner.

## The influence of microglia replacement on the general anesthesia

In 2020, we first developed efficient strategies for microglia replacement and proposed therapeutic applications for neurological disorders (*Xu et al., 2020*). Microglia replacement by bone marrow transplantation (Mr BMT or mrBMT), one of the replacement strategies, can induce bone marrow cells (BMCs) to differentiate into microglia-like cells and efficiently replace endogenous microglia in the whole CNS (*Xu et al., 2020*; *Xu et al., 2021c*). Despite sharing similar characteristics to endogenous microglia, the replaced cells are P2Y12$^-$ (*Figure 13A and B*) as we previously reported (*Xu et al., 2020*). We thus reasoned that if microglial P2Y12 indeed influences the response to general anesthesia, Mr BMT-treated mice with P2Y12$^-$ microglia should display a dampened response to anesthetics. To this end, we examined anesthesia induction and emergence in Mr BMT mice (*Figure 13A*). We found that Mr BMT mice exhibited delayed LORR and early RORR (*Figure 13C*), further echoing the important regulatory role of P2Y12 in the response to general anesthesia.

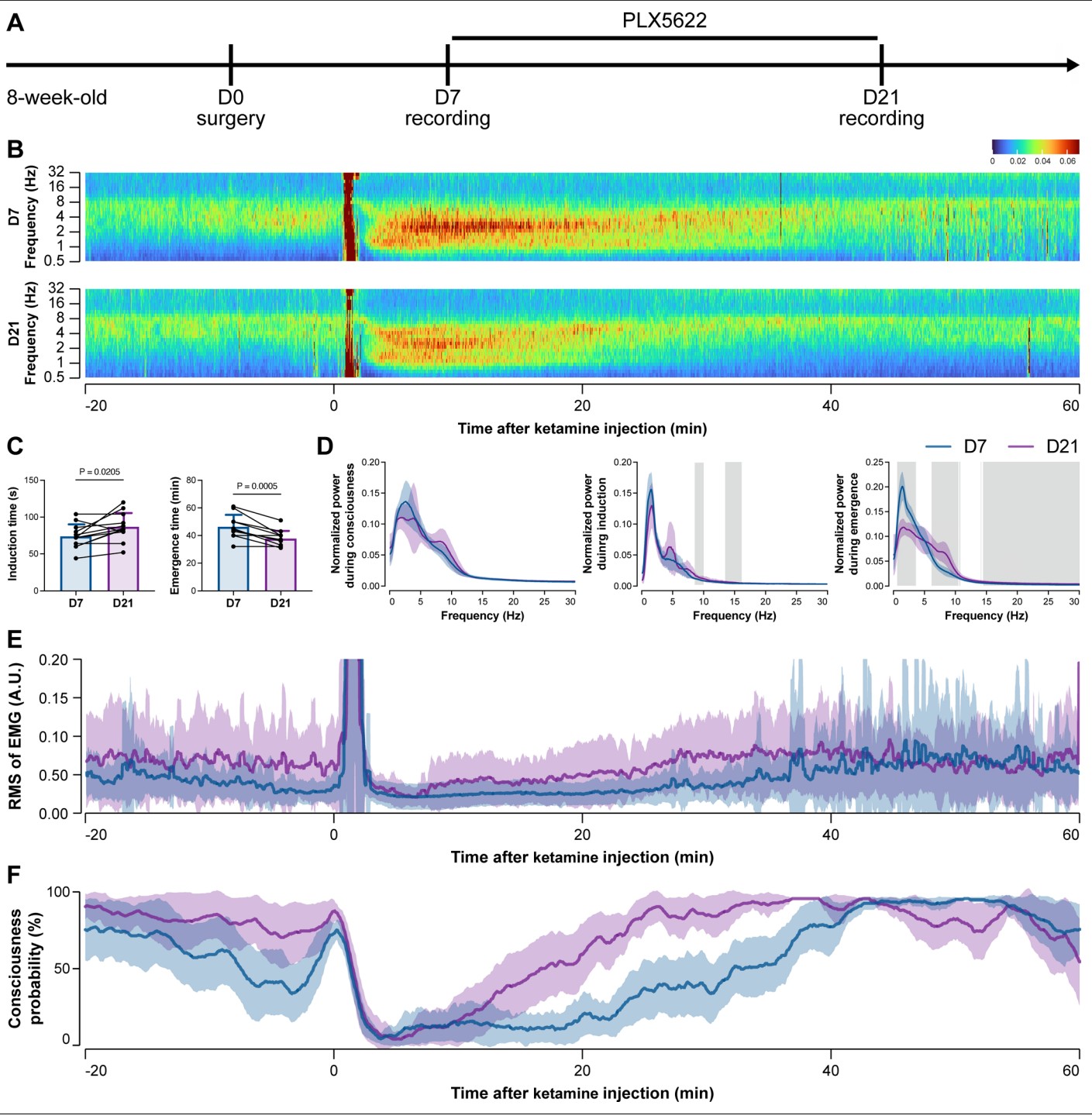

**Figure 6.** Electroencephalography (EEG) and electromyography (EMG) recordings reveal that mice with microglial depletion are mouse resistant to general anesthesia by ketamine. (**A**) Scheme of time points for animal surgery, microglial depletion, and EEG/EMG recording. (**B–D**) Microglial depletion does not change the EEG before the injection of ketamine. Instead, it influences the EEG in anesthesia induction and emergence. Two-tailed paired t-test. The gray area in (**D**) indicates p<0.05 between D7 and D21. (**E**) Microglial depletion does not change the EMG before the injection of ketamine. Instead, it influences the EMG in the anesthesia process. (**F**) Microglial depletion does not change the probability of consciousness before the injection of ketamine. Instead, it influences the consciousness probability in the anesthesia process. N = 12mice for each group. Data are presented as mean ± SD. RMS: root mean square; A.U.: arbitrary unit; PLX5622: PLX5622-formulated diet. All animals are male mice.

The online version of this article includes the following source data for figure 6:

**Source data 1.** Raw data for induction and emergence times.

*Figure 6 continued on next page*

*Figure 6 continued*

**Source data 2.** Raw data for the normalized power during different stages.

**Source data 3.** Raw data for the RMS of EMG.

**Source data 4.** Raw data for the consciousness probability.

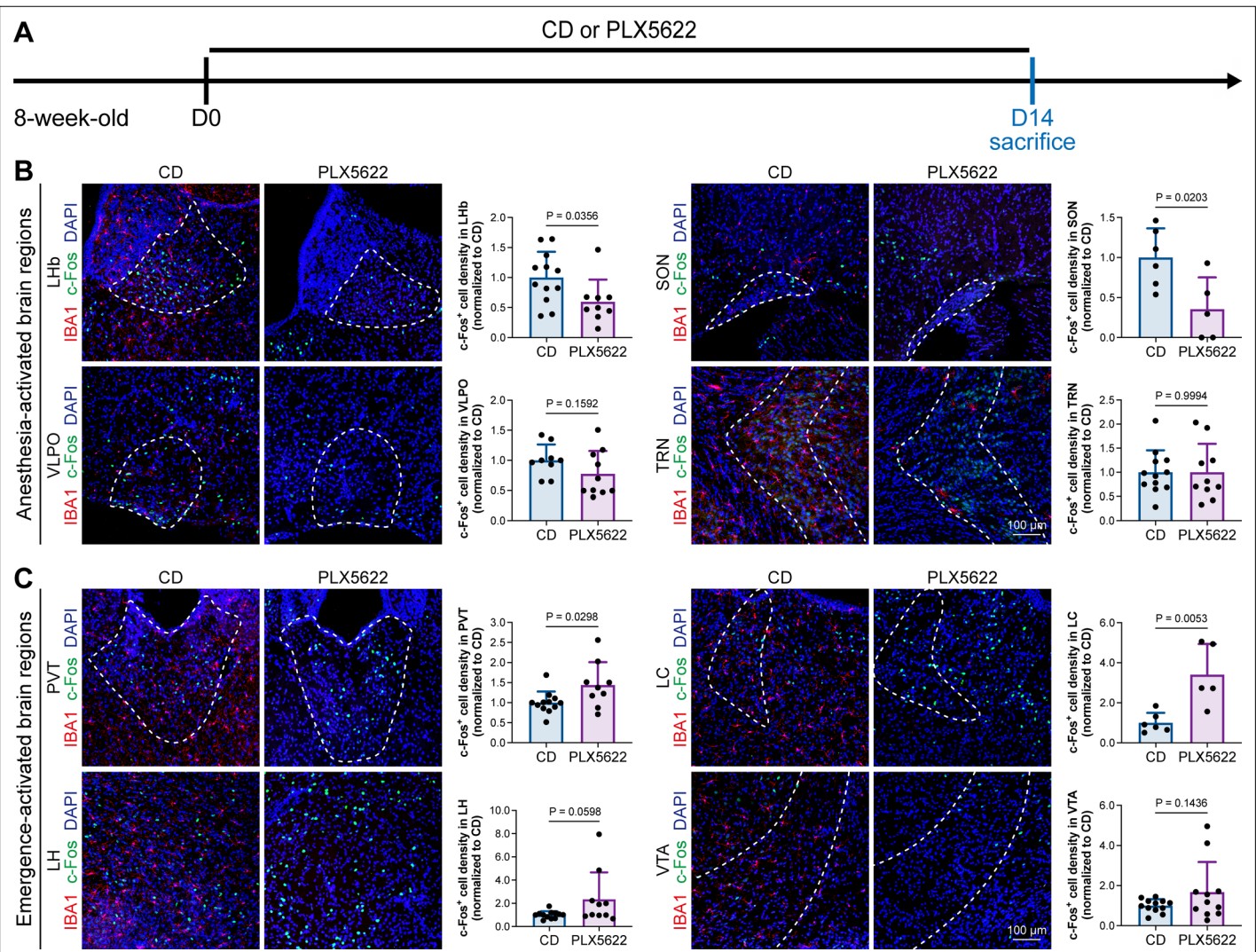

**Figure 7.** Microglial depletion diversely influences neuronal activity in different anesthesia-related brain regions. (**A**) Scheme of time points for microglial depletion and examination time points. (**B**) Influence of microglial depletion in anesthesia-activated brain regions. Microglial depletion reduces neuronal activity in the LHb (p=0.0356), SON (p=0.0203), and VLPO (p=0.1592) and does not influence neuronal activity in the TRN (p=0.9994). N = 12, 6, 9, and 12mice for LHb, SON, VLPO, and TRN in the CD group, respectively. N = 9, 5, 10, and 10mice for LHb, SON, VLPO, and TRN in the PLX5622 group, respectively. (**C**) Influence of microglial depletion in emergence-activated brain regions. Microglial depletion enhances neuronal activity in the PVT (p=0.0298), LC (p=0.0053), LH (p=0.0598), and VTA (p=0.1436). N = 12, 6, 12, and 12mice for PVT, LC, LH, and VTA in the CD group, respectively. N = 9, 5, 10, and 11mice for PVT, LC, LH, and VTA in the PLX5622 group, respectively. Two-tailed independent *t*-test. Data are presented as mean ± SD. PLX5622: PLX5622-formulated diet; CD: control diet; LHb: lateral habenula; SON: supraoptic nucleus; VLPO: ventrolateral preoptic nucleus; TRN: thalamic reticular nucleus; PVT: paraventricular thalamus; LC: locus coeruleus; LH: lateral hypothalamus; VTA: ventral tegmental area. All animals are male mice.

The online version of this article includes the following source data for figure 7:

**Source data 1.** The influence of microglial depletion to the c-Fos⁺ cell density in the AABRs and EABRs.

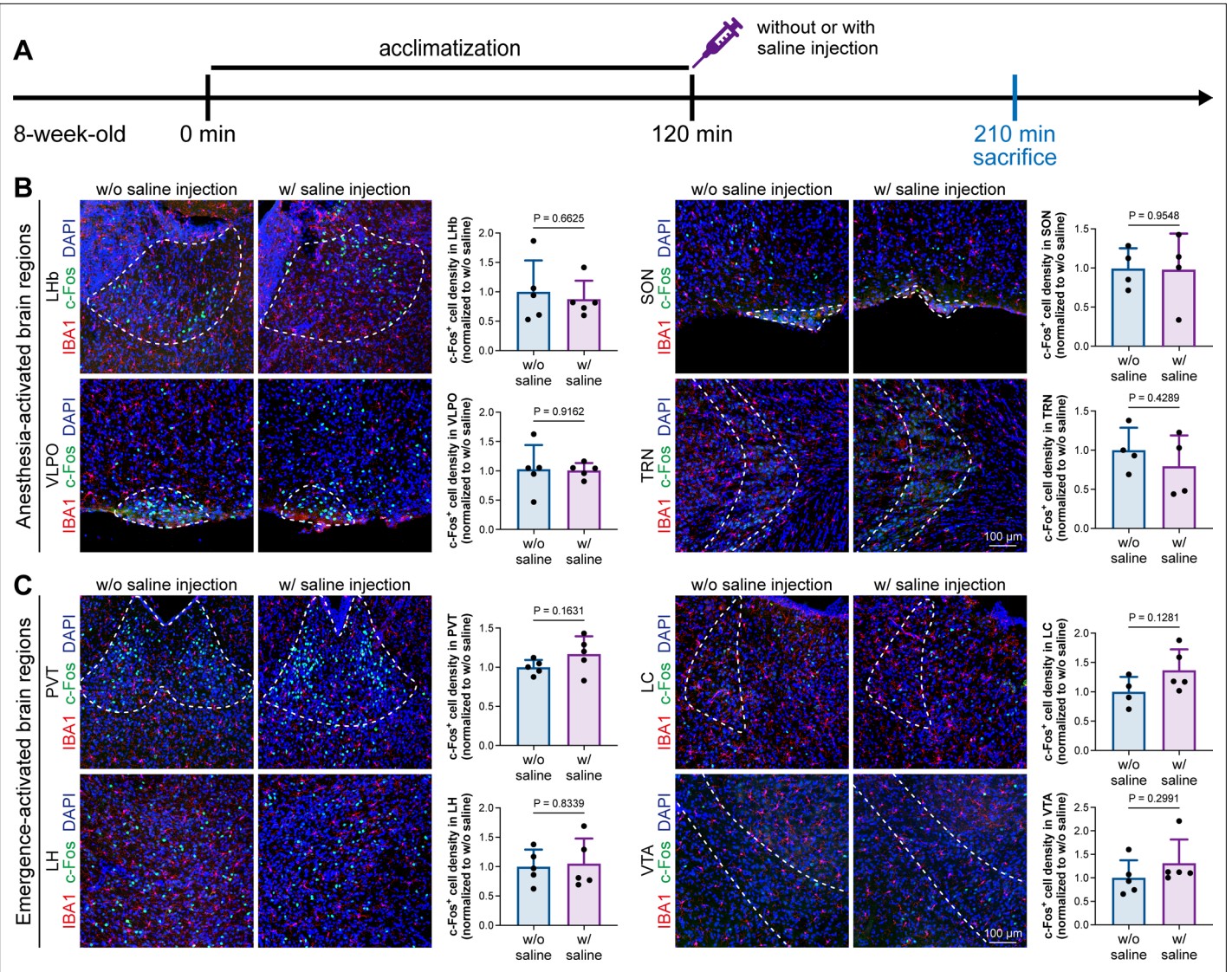

**Figure 8.** Animal handling and intraperitoneal injection do not influence neuronal activity in anesthesia-related brain regions. (**A**) Scheme of time points for microglial depletion and examination time points. (**B**) Animal handling and intraperitoneal injection do not influence neuronal activity in the LHb, SON, VLPO, or TRN. N = 5, 4, 5, and 4mice for LHb, SON, VLPO, and TRN in the w/o saline group, respectively. N = 5, 4, 5, and 4mice for LHb, SON, VLPO, and TRN in the w/ saline group, respectively. (**C**) Animal handling and intraperitoneal injection do not influence neuronal activity in the PVT, LC, LH, or VTA. N = 5, 4, 5, and 5mice for PVT, LC, LH, and VTA in the w/o saline group, respectively. N = 5, 5, 5, and 5mice for PVT, LC, LH, and VTA in the w/ saline group, respectively. Two-tailed independent *t*-test. Data are presented as mean ± SD. LHb: lateral habenula; SON: supraoptic nucleus; VLPO: ventrolateral preoptic nucleus; TRN: thalamic reticular nucleus; PVT: paraventricular thalamus; LC: locus coeruleus; LH: lateral hypothalamus; VTA: ventral tegmental area. All animals are male mice.

The online version of this article includes the following source data for figure 8:

**Source data 1.** Influences of the animal handling and intraperitoneal injection to the c-Fos+ cell density in the AABRs and EABRs.

## Microglial intracellular calcium regulates general anesthesia

Purinergic activation of P2Y12 increases intracellular $Ca^{2+}$ (*Jiang et al., 2017*; *Jairaman et al., 2022*; *Pozner et al., 2015*). We thus reasoned that the modulation of the anesthesia response is mediated by downstream $Ca^{2+}$ signaling. To this end, we ectopically expressed a chemogenetic receptor in microglia by *Cx3cr1*[+/CreER]::hM3Dq-YFP[+/−] (*Figure 14A and B*). hM3Dq is a modified human M3 muscarinic (hM3) receptor that activates $G_{αq}$ upon clozapine-N-oxide (CNO) administration and in turn enhances its downstream $Ca^{2+}$ concentration (*Roth, 2016*; *Urban and Roth, 2015*). When we treated *Cx3cr1*[+/CreER]::hM3Dq-YFP[+/−] with CNO to elevate the intracellular $Ca^{2+}$ level in microglia, the LORR to

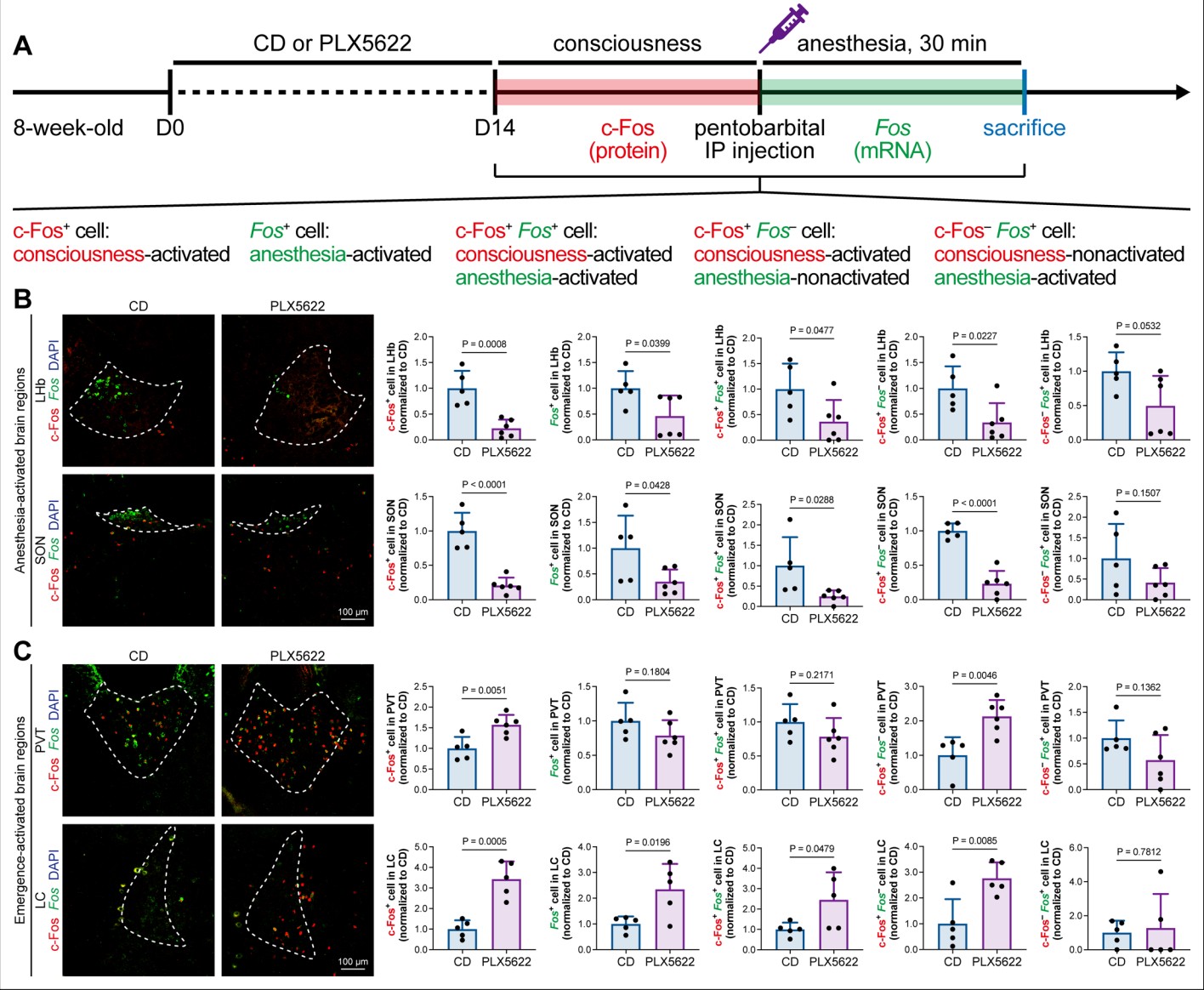

**Figure 9.** c-Fos protein and *Fos* mRNA dual staining dissects the influence of microglial depletion on consciousness and anesthesia states. (**A**) Scheme of time points for microglial depletion and dual labeling. (**B, C**) The influence of microglial depletion on activated neurons in consciousness and anesthesia states in AABRs (LHb and SON) and EABRs (PVT and LC). N = 5 (LHb CD), 6 (LHb PLX5622), 5 (SON CD), 6 (SON PLX5622), 5 (PVT CD), 6 (PVT PLX5622), 5 (LC CD), and 5 (LC PLX5622) mice for each group. Two-tailed independent *t*-test. Data are presented as mean ± SD. PLX5622: PLX5622-formulated diet; AABRs; anesthesia-activated brain regions; EABRs; emergence-activated brain regions; CD: control diet; LHb: lateral habenula; SON: supraoptic nucleus; PVT: paraventricular thalamus; LC: locus coeruleus. All animals are male mice.

The online version of this article includes the following source data for figure 9:

**Source data 1.** Influences of microglial depletion to the c-Fos[+], Fos[+], c-Fos[+]Fos[+], c-Fos[+]Fos[-], and c-Fos[-]Fos[+] cells in the AABRs and EABRs.

pentobarbital was accelerated, and RORR was delayed (*Figure 14C*). On the other hand, STIM1 is an endoplasmic reticulum $Ca^{2+}$ sensor. The lack of STIM1 results in impaired store-operated $Ca^{2+}$ influx (*Oh-Hora et al., 2008*; *Zhang et al., 2005*; *Stiber et al., 2008*). To specifically disrupt intracellular $Ca^{2+}$ signaling in microglia, we conditionally knocked out STIM1 in microglia in *Cx3cr1*[+/CreER]::*Stim1*[fl/fl] mice. After tamoxifen induction, *Stim1* mRNA was significantly reduced in the *Cx3cr1*[+/CreER]::*Stim1*[fl/fl] mouse brain (*Figure 14D and E*). We found that with impaired $Ca^{2+}$ signaling in microglia, *Cx3cr1*[+/CreER]::*Stim1*[fl/fl] mice displayed delayed anesthesia induction and early emergence (*Figure 14F*). By both enhancing and disrupting microglial $Ca^{2+}$, our results reveal that intracellular $Ca^{2+}$ in microglia facilitates the anesthesia process.

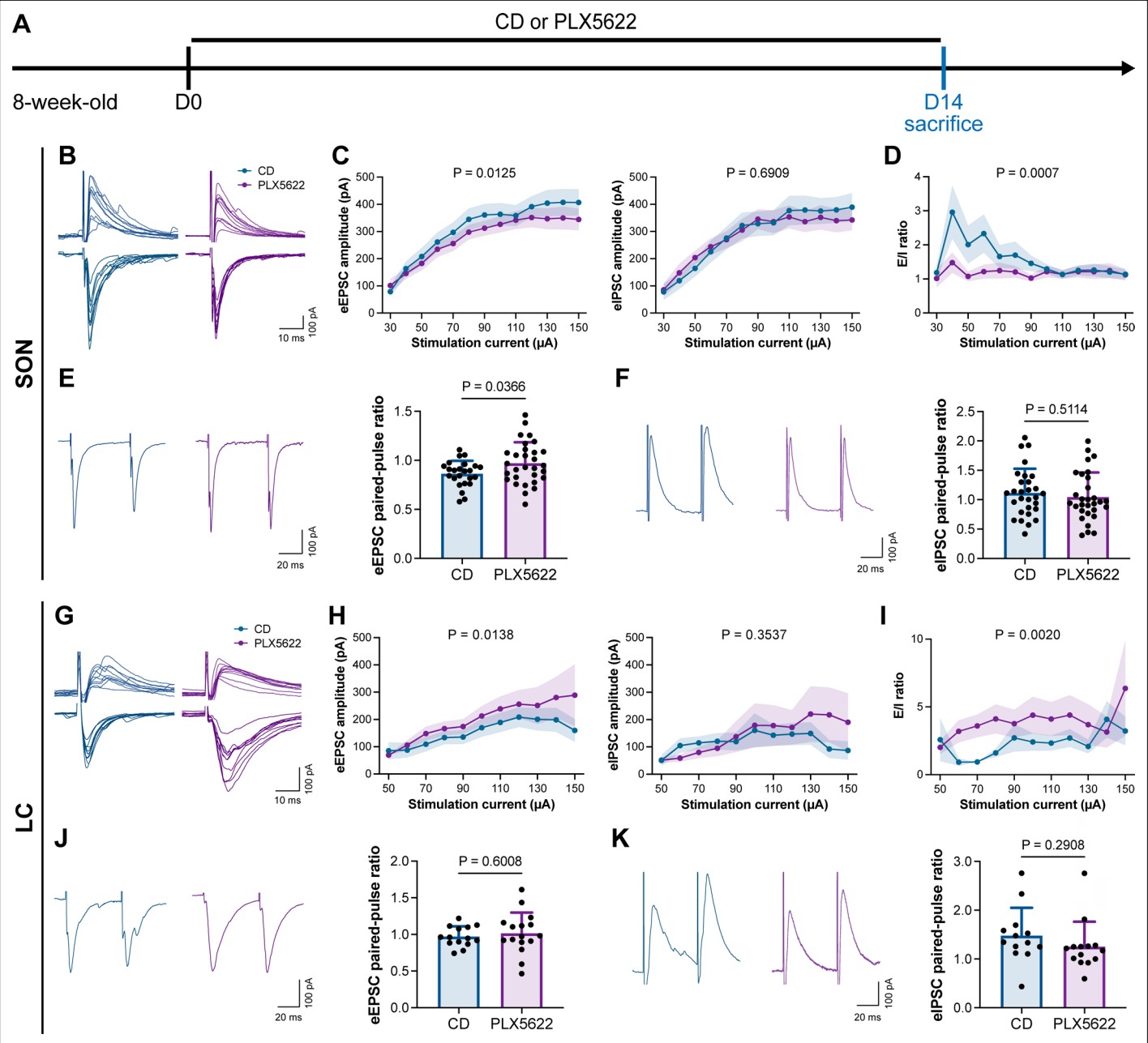

**Figure 10.** Microglial depletion reduces the E/I ratio in SON but enhances the E/I ratio in LC. (**A**) Scheme of time points for microglial depletion by PLX5622. (**B**) Representative traces for evoked postsynaptic currents in the SON to 10 increasing stimulation currents. (**C**) Amplitudes of evoked postsynaptic currents in the SON in response to increasing electrical stimulation intensities. Two-way ANOVA. Data are presented as mean ± SEM. (**D**) E/I ratios with different stimulation intensities in the SON. N = 21 (CD) and 19 (PLX5622) cells from fivemice for each group. Two-way ANOVA. Data are presented as mean ± SEM. (**E**) Representative traces (left) and quantitative results (right) show that PLX5622-treated mice exhibited a higher eEPSC PPR in the SON. N = 24 (CD) and 30 (PLX5622) cells from fivemice for each group. Two-tailed independent *t*-test. Data are presented as mean ± SD. (**F**) Representative traces (left) and quantitative results (right) show that PLX5622-treated mice exhibited a similar eIPSC PPR in the SON. N = 29 (CD) and 30 (PLX5622) cells from fivemice for each group. Two-tailed independent *t*-test. Data are presented as mean ± SD. (**G**) Representative traces for evoked postsynaptic currents in the LC in response to 10 increasing stimulation currents. (**H**) Amplitudes of evoked postsynaptic currents in the LC in response to increasing electrical stimulation intensities. in response to the electrical stimulation. N = 15 (EPSC CD), 18 (EPSC PLX5622), 15 (IPSC CD), and 18 (IPSC PLX5622) cells from fivemice for each group. Two-way ANOVA. Data are presented as mean ± SEM. (**I**) E/I ratios with different stimulation currents in the LC. N = 15 (EPSC CD), 18 (EPSC PLX5622), 15 (IPSC CD), and 18 (IPSC PLX5622) cells from fivemice for each group. Two-way ANOVA. Data are presented as mean ± SEM. (**J**) Representative traces (left) and quantitative results (right) show that PLX5622-treated mice exhibited a similar eEPSC PPR in the LC. N = 14 (CD) and 16 (PLX5622) cells from fivemice for each group. Two-tailed independent *t*-test. Data are presented as mean ± SD. (**K**) Representative traces (left) and quantitative results (right) show that PLX5622-treated mice exhibited a similar eIPSC PPR in the LC. N = 13 (CD) and

*Figure 10 continued on next page*

*Figure 10 continued*

14 (PLX5622) cells from fivemice for each group. Two-tailed independent *t*-test. Data are presented as mean ± SD. PLX5622: PLX5622-formulated diet; SON: supraoptic nucleus; LC: locus coeruleus; PPR: paired-pulse ratio (PPR); CD: control diet. eEPSC: evoked excitatory postsynaptic current; eIPSC: evoked inhibitory postsynaptic current. All animals are male mice.

The online version of this article includes the following source data for figure 10:

**Source data 1.** Raw data for eEPSC/eIPSC amplitudes and E/I ratios in the SON.

**Source data 2.** Raw data for eEPSC/eIPSC PPRs in the SON.

**Source data 3.** Raw data for eEPSC/eIPSC amplitudes and E/I ratios in the LC.

**Source data 4.** Raw data for eEPSC/eIPSC PPRs in the LC.

Purinergic activation of P2Y12 enhances intracellular $Ca^{2+}$ (*Jiang et al., 2017*; *Jairaman et al., 2022*; *Pozner et al., 2015*). Our results thus indicate that microglia regulate the anesthesia process through P2Y12 and its downstream $Ca^{2+}$ signaling.

## Discussion

### The mutual interaction between microglia and neurons

Previous studies have indicated that microglia exhibit increased process motility, extension, and territory surveillance during anesthetization and sleep (*Liu et al., 2019b*; *Stowell et al., 2019*). However, whether and how microglia regulate neuronal activity and contribute to anesthesia response is largely unknown. Our study demonstrated an active role of microglia in neuronal activity that facilitates and stabilizes the anesthesia response by differentially changing neuronal activity in the AABRs and EABRs. It relies on microglial P2Y12 and intracellular calcium, rather than the spines' plasticity (*Figure 15*). Microglia and neurons mutually interact with each other under both physiological and pathological conditions. Previous studies have indicated that neurons can influence the morphology and function of microglia through neurotransmitters and/or neuromodulators, such as GABA and ATP (*Logiacco et al., 2021*; *Dissing-Olesen et al., 2014*). On the other hand, microglia can regulate neuronal activity. Microglia in the paraventricular nucleus are able to maintain the balance of sympathetic outflow and suppress the pressor response under hypertensive insults (*Bi et al., 2022*). Chemogenetic manipulations of microglia lead to a prostaglandin-dependent reduction in the excitability of striatal neurons (*Klawonn et al., 2021*). This evidence reveals that although microglia are resident immune cells in the brain, their functions are not limited to the immune response. Our study found that during the process of general anesthesia, microglia serve as an 'anesthesia facilitator and stabilizer' by activating AABRs and inhibiting EABRs. As a result, microglia-depleted mice are more resistant to general anesthesia.

### Microglial depletion diversely influences neuronal activities in different brain regions

Different nuclei are involved in the response to general anesthesia. The influences of microglial depletion on neuronal activity among these nuclei are different. After microglial ablation, c-Fos expression is decreased in AABRs but increased in EABRs. Meanwhile, the electrophysiology results also show that the E/I ratio is differentially regulated in different brain regions upon microglial depletion. However, the mechanism behind brain region-specific regulation is unclear. Several hypotheses may explain the microglia-mediated diverse regulations among different brain regions. First, it may be due to microglial heterogeneity among brain regions. Nonetheless, a recent study indicated that the cross-regional heterogeneity in adulthood was overestimated in previous studies (*Li et al., 2019*). Second, different neuronal cell types may differentially respond to microglial depletion. Take the adenosine receptor as an example. The adenosine concentration is reduced in the microglia-depleted cortex (*Császár et al., 2022*). Interference of P2Y12, CD39, or CD73 in microglia disrupts the metabolism of extracellular adenosine in the brain (*Badimon et al., 2020*). Our results showed that inhibition or knockout of P2Y12 results in resistance of general anesthesia, indicating that neurons in different brain regions, for example, the AABRs and EABRs, differentially respond to adenosine. Moreover, adenosine receptor subtypes are discriminately distributed across different brain regions (*Liu et al., 2019a*), suggesting that neurons of different adenosine receptors in the different brain regions may differentially respond to adenosine. Third, the brain region-specific regulation may also rely on the neural circuitry. Mutual

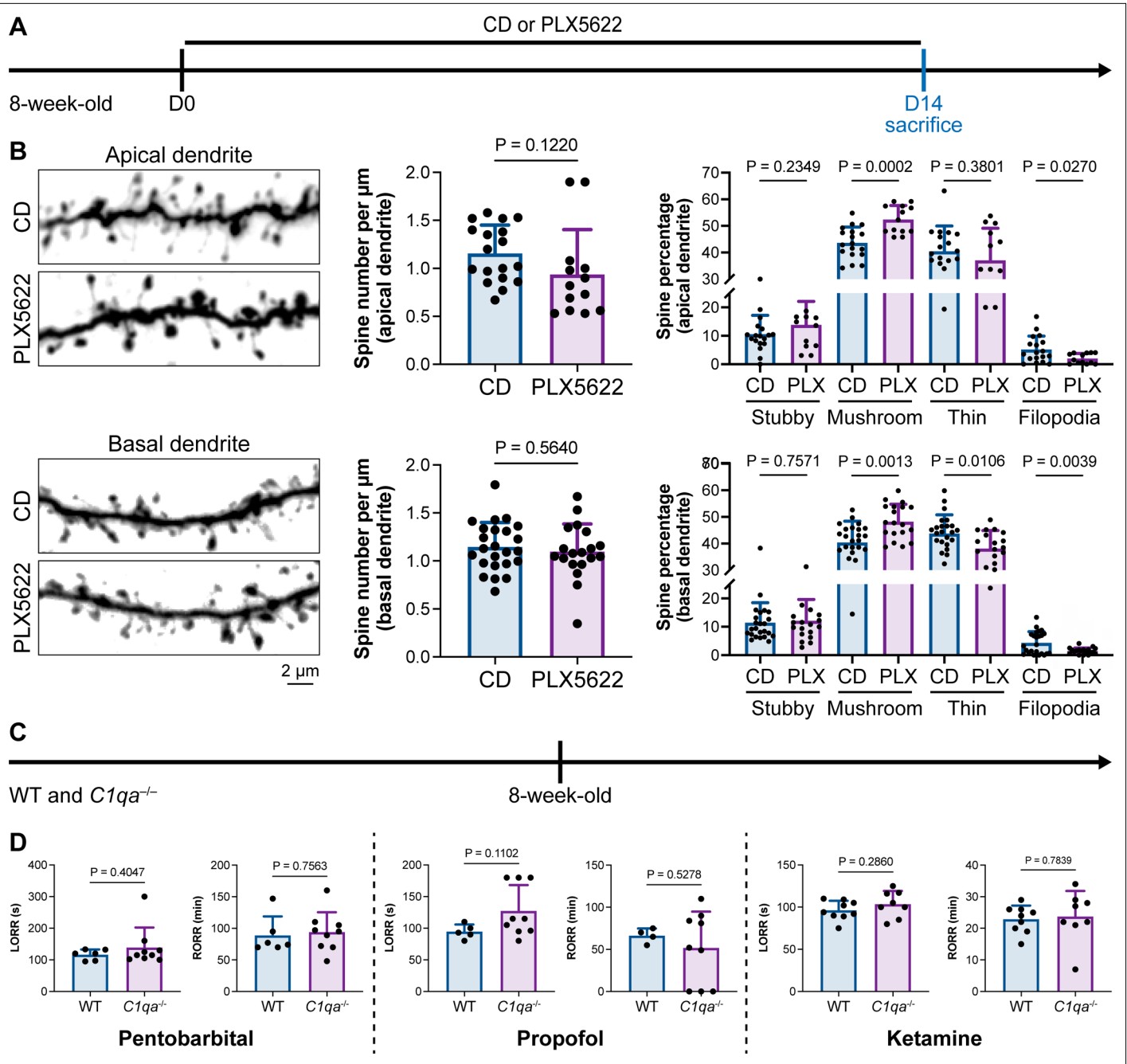

**Figure 11.** Interruption of the spine 'eat me' signal by *C1qa*[−/−] does not influence the anesthesia process and microglial depletion alters the proportion of spine categories. (**A**) Scheme of time points for microglial depletion and examination time points. (**B**) CSF1R inhibition for 14d does not influence spine density but changes the proportion of spine subtypes. N = 18and 13cells from fivemice for each group of apical spines, N = 24and 19cells from fivemice for each group of basal spines. All animals are male mice. (**C**) Scheme of LORR and RORR tests in wild-type and *C1qa*[−/−] mice. (**D**) C1q knockout does not influence anesthesia induction and emergence in response to pentobarbital, propofol, and ketamine. N = 6 (pentobarbital WT; five male and one female mice), 9 (pentobarbital *C1qa*[−/−]; seven male and two female mice), 5 (propofol WT; four male and one female mice), 9 (propofol *C1qa*[−/−]; seven male and two female mice), 9 (ketamine WT; nine male mice), and 8 (ketamine *C1qa*[−/−]; eight male mice). Both sexes are used in this result. Two-tailed independent *t*-test. Data are presented as mean ± SD. PLX5622: PLX5622-formulated diet; CD: control diet; LORR: loss of righting reflex; RORR: recovery of righting reflex.

The online version of this article includes the following source data for figure 11:

**Source data 1.** Raw data for the spine density and the proportion of each spine subtype.

**Source data 2.** Raw data for LORR and RORR times.

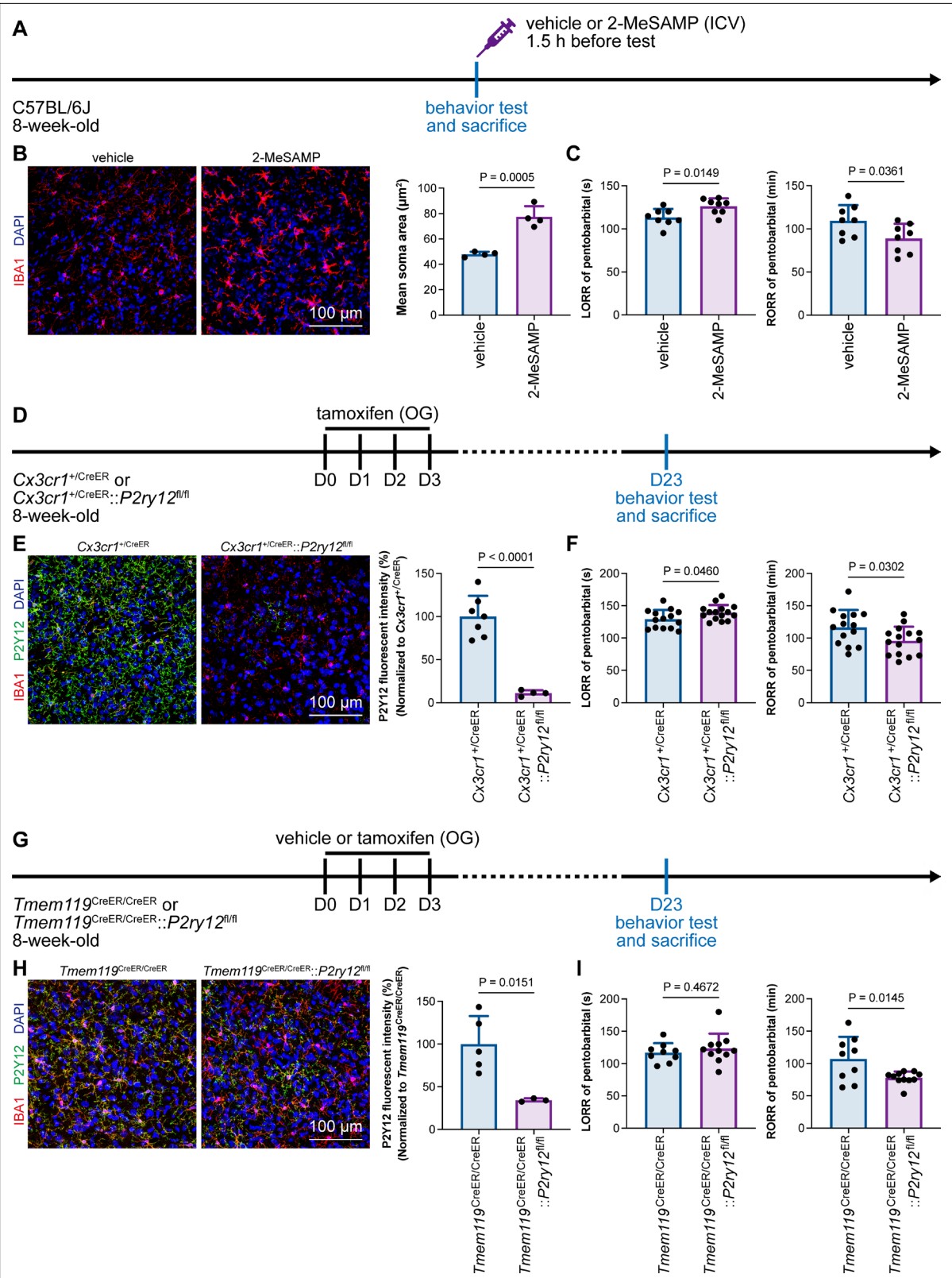

**Figure 12.** Microglial P2Y12 regulates the induction and emergence of anesthesia. (**A**) Scheme of 2-MeSAMP administration and behavior tests for anesthesia. (**B**) P2Y12 inhibition by 2-MeSAMP drives microglia to a more reactive state. N = 4 mice for each group. All animals are male mice. (**C**) P2Y12 inhibition by 2-MeSAMP results in delayed anesthesia induction and early emergence. N = 8 mice for each group. All animals are male mice. (**D**) Scheme of animal treatment and examination time points for *Cx3cr1*[+/CreER] and *Cx3cr1*[+/CreER]::*P2ry12*[fl/fl] mice. (**E**) Tamoxifen induces efficient P2Y12 knockout in

*Figure 12 continued on next page*

*Figure 12 continued*

$Cx3cr1^{+/CreER}$::$P2ry12^{fl/fl}$ mice. N = 7mice for the $Cx3cr1^{+/CreER}$ group and 4mice for the $Cx3cr1^{+/CreER}$::$P2ry12^{fl/fl}$ group. All animals are male mice. (**F**) Efficient knockout of P2Y12 significantly elongates the LORR and shortens the RORR. N = 14mice (eleven male and three female mice) for the $Cx3cr1^{+/CreER}$ group and 15mice (eleven male and four female mice) for the $Cx3cr1^{+/CreER}$::$P2ry12^{fl/fl}$ group. (**G**) Scheme of animal treatment and examination time points for $Tmem119^{CreER/CreER}$ and $Tmem119^{CreER/CreER}$::$P2ry12^{fl/fl}$ mice. (**H**) Tamoxifen induces relatively low efficiency of P2Y12 knockout in $Tmem119^{CreER/CreER}$::$P2ry12^{fl/fl}$ mice. N = 5mice for the $Tmem119^{CreER/CreER}$ group and 3mice for the $Tmem119^{CreER/CreER}$::$P2ry12^{fl/fl}$ group. All animals are male mice. (**I**) Low-efficiency knockout of P2Y12 does not affect anesthesia induction but significantly shortens the emergence time. N = 9mice (six male and three female mice) for the $Tmem119^{CreER/CreER}$ group and 11mice (seven male and four female mice) for the $Tmem119^{CreER/CreER}$::$P2ry12^{fl/fl}$ group. Two-tailed independent $t$-test. Data are presented as mean ± SD. ICV: intracerebroventricular; OG: oral gavage; LORR: loss of righting reflex; RORR: recovery of righting reflex.

The online version of this article includes the following source data for figure 12:

**Source data 1.** Influences of 2-MeSAMP treatment to the microglial morphology and LORR and RORR times.

**Source data 2.** Knockout efficiency of $Cx3cr1^{+/CreER}$::$P2ry12^{fl/fl}$ mice and the influence of high-efficiency P2Y12 knockout to LORR and RORR times.

**Source data 3.** Knockout efficiency of $Tmem119^{CreER/CreER}$::$P2ry12^{fl/fl}$ mice and the influence of low-efficiency P2Y12 knockout to LORR and RORR times.

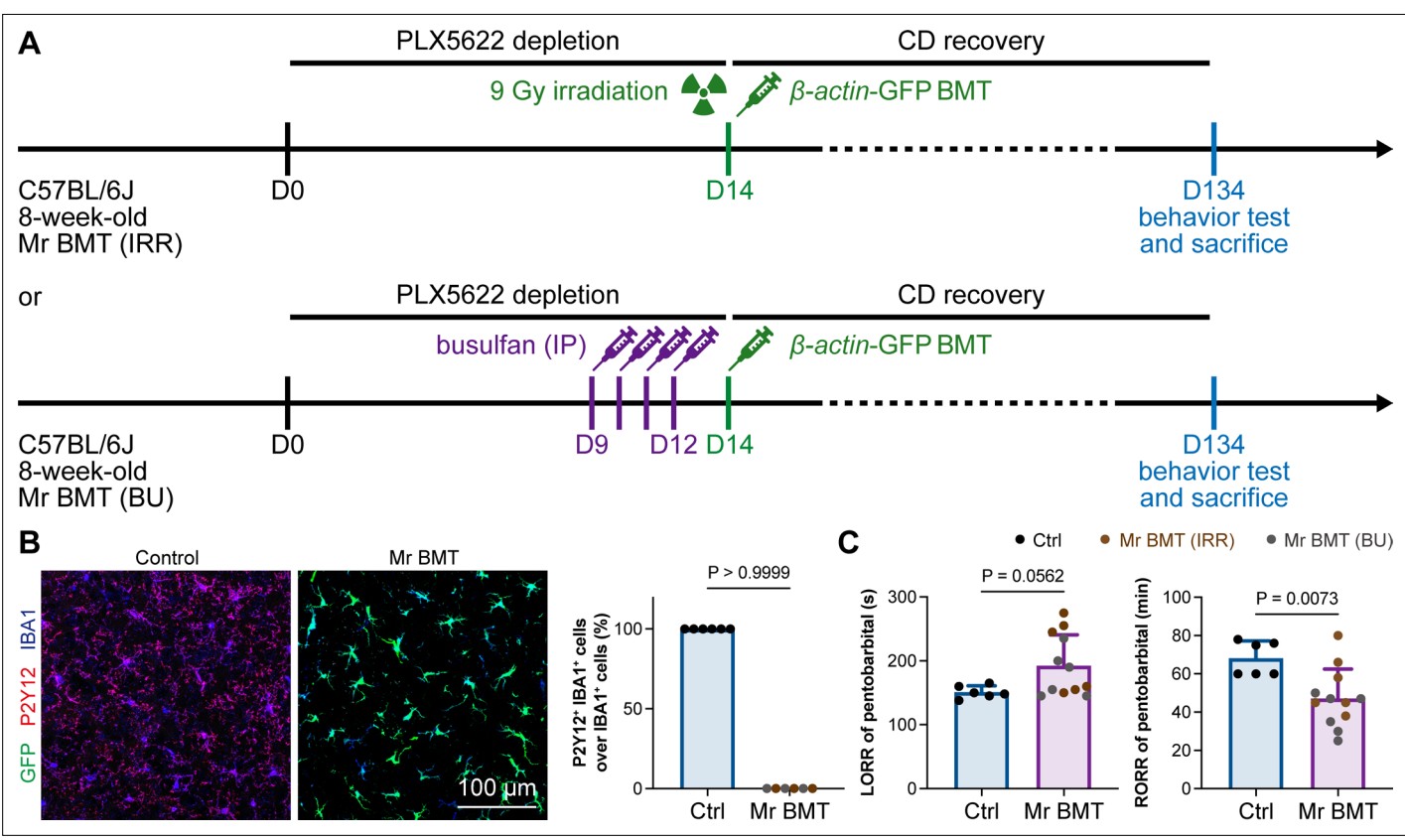

**Figure 13.** Mice with P2Y12⁻ Mr BMT cells display delayed anesthesia induction and early emergence. (**A**) Scheme of microglia replacement by Mr BMT and behavior tests for anesthesia. (**B**) Mr BMT cells exhibit a P2Y12⁻ phenotype. N = 6mice for each group. (**C**) P2Y12⁻ microglia lead to delayed anesthesia induction and early emergence. N = 6mice for the control group, 6mice for the Mr BMT (IRR) group, and 6mice for the Mr BMT (BU) group. Two-tailed independent $t$-test. Data are presented as mean ± SD. IP: intraperitoneal injection; Mr BMT: microglia replacement by bone marrow transplantation; BMT: bone marrow transplantation; Ctrl: control; IRR: irradiation; BU: busulfan; LORR: loss of righting reflex; RORR: recovery of righting reflex. All animals are female mice.

The online version of this article includes the following source data for figure 13:

**Source data 1.** Percentages of P2Y12⁺ IBA1⁺ cells over IBA1⁺ cells.

**Source data 2.** Raw data for LORR and RORR times.

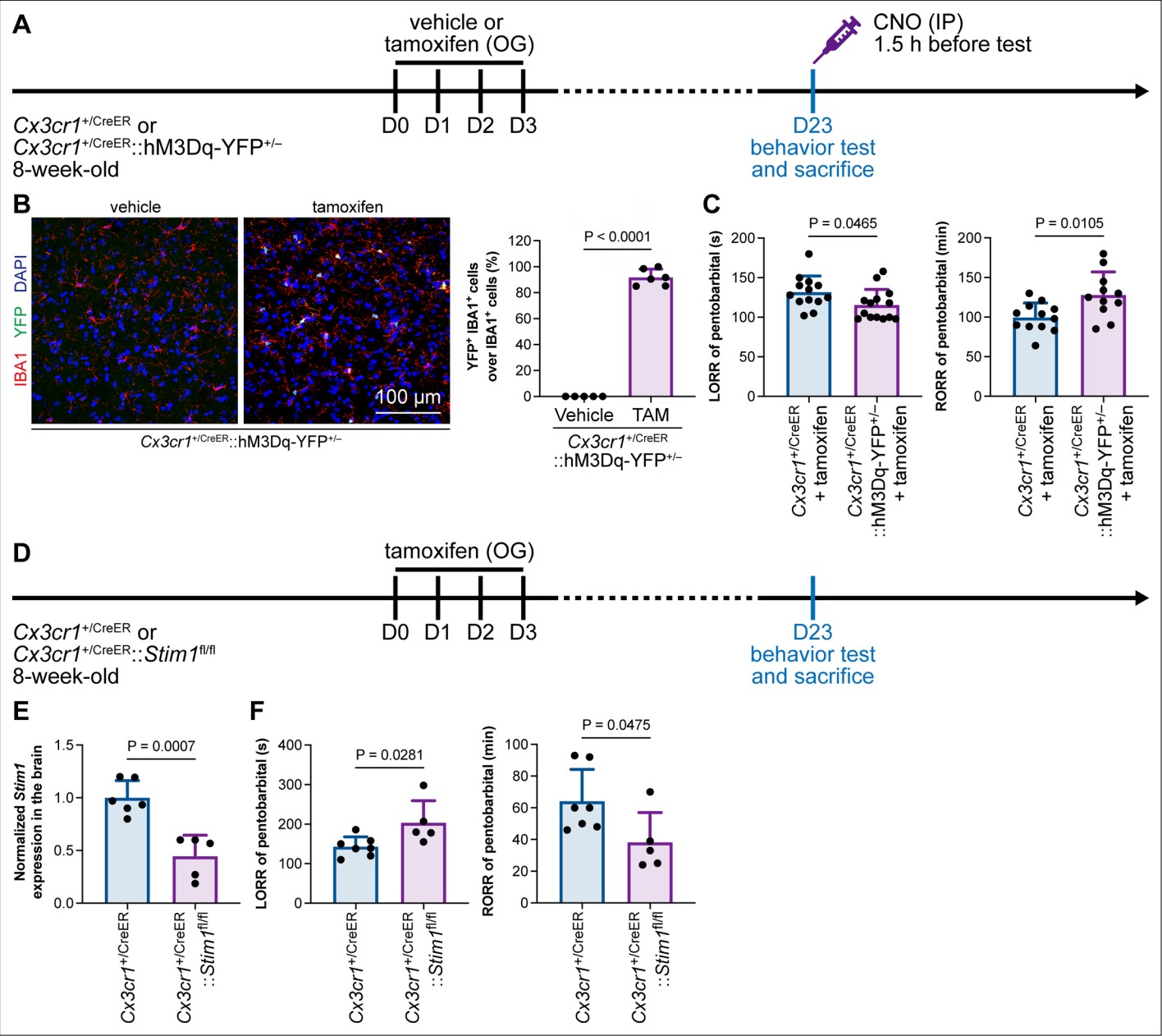

**Figure 14.** General anesthesia is regulated by intracellular calcium in microglia. (**A**) Scheme of animal treatment and examination time points for *Cx3cr1*[+/CreER] and *Cx3cr1*[+/CreER]::hM3Dq-YFP[+/−] mice. (**B**) Tamoxifen induces high Cre-dependent recombination in *Cx3cr1*[+/CreER]::hM3Dq-YFP[+/−] mice. N = 5mice for the vehicle group and 6mice for the tamoxifen group. All animals are male mice. (**C**) Elevation of microglial intracellular Ca[2+] results in a shorter anesthesia induction time and longer emergence time. N = 13 (LORR *Cx3cr1*[+/CreER]; eleven male and two female mice), 14 (LORR *Cx3cr1*[+/CreER]::hM3Dq-YFP[+/−]; ten male and four female mice), 12 (RORR *Cx3cr1*[+/CreER]; ten male and two female mice), and 11mice (RORR *Cx3cr1*[+/CreER]::hM3Dq-YFP[+/−]; seven male and 4 female mice) per group. (**D**) Scheme of animal treatment and examination time points for *Cx3cr1*[+/CreER] and *Cx3cr1*[+/CreER]::*Stim1*[fl/fl] mice. (**E**) qPCR results reveal decreased *Stim1* transcription in *Cx3cr1*[+/CreER]::*Stim1*[fl/fl] mouse brains. N = 6mice (five male and one female mice) for the *Cx3cr1*[+/CreER] group and 5mice (three male and two female mice) for the *Cx3cr1*[+/CreER]::*Stim1*[fl/fl] group. (**F**) Downregulation of microglial intracellular Ca[2+] results in longer anesthesia induction time and shorter emergence time. N = 7 *Cx3cr1*[+/CreER] (five male and two female mice) and 5 *Cx3cr1*[+/CreER]::*Stim1*[fl/fl] mice (three male and two female mice) per group. Two-tailed independent *t*-test. Data are presented as mean ± SD. OG: oral gavage; IP: intraperitoneal injection; TAM: tamoxifen; LORR: loss of righting reflex; RORR: recovery of righting reflex.

The online version of this article includes the following source data for figure 14:

**Source data 1.** Efficiency of Cre-dependent recombination in *Cx3cr1*[+/CreER]::hM3Dq-YFP[+/−] mice and the influence of Ca[2+]-elevated microglia to LORR and RORR times.

**Source data 2.** Assessment of Stim1 knockout efficiency in *Cx3cr1*[+/CreER]::*Stim1*[fl/fl] mice and the influence of Ca[2+]-reduced microglia to LORR and RORR times.

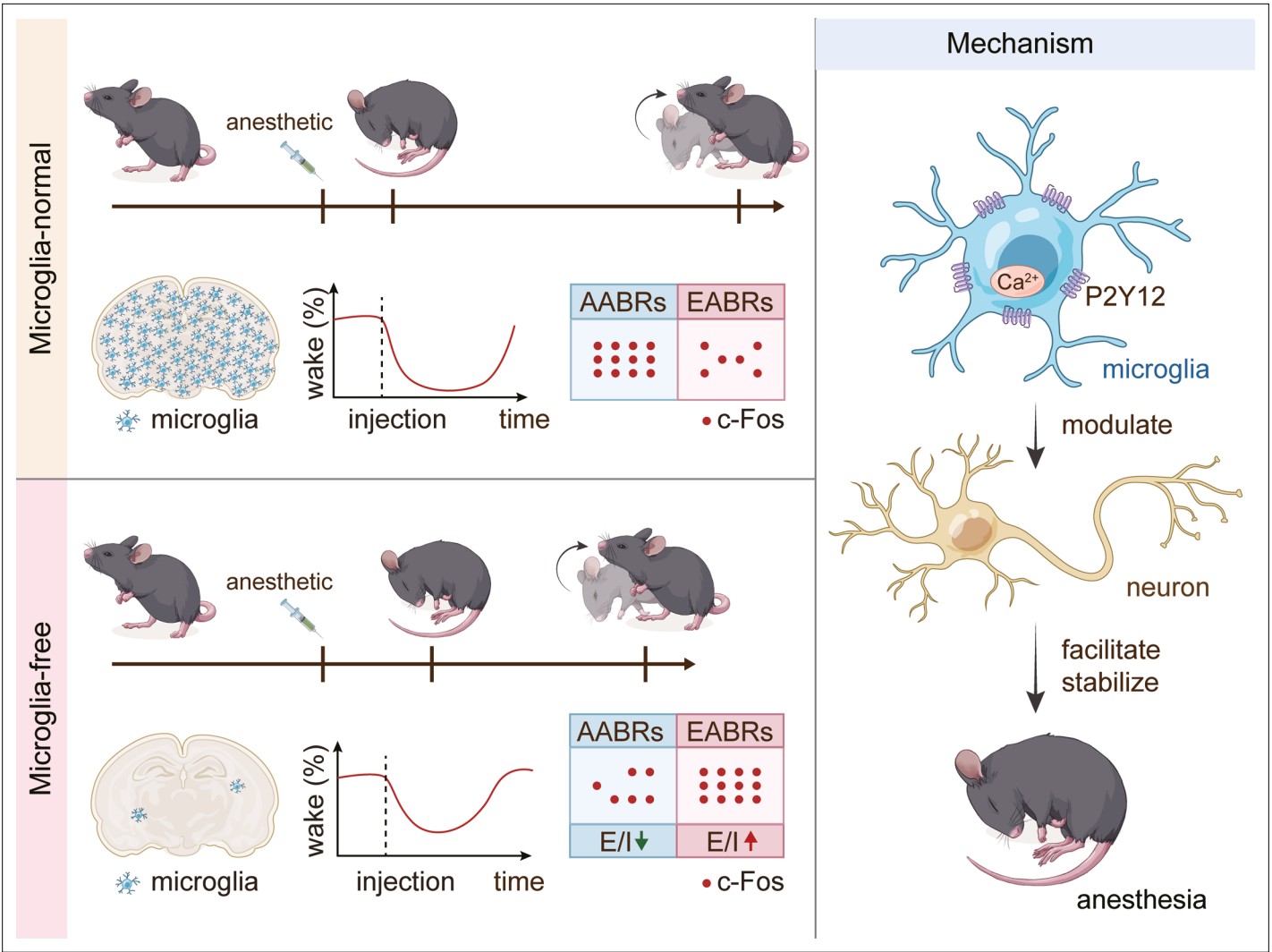

**Figure 15.** Schematic summary of this study. This figure summarizes the major findings of this study.

innervations of the AABRs-EABRs and local circuitries within specific brain region can result in the diverse neuronal response. The regulation of neuronal activity is an overall effect that integrated with multiple variables. So does the microglial contribution to the different brain regions.

## Molecular mechanisms of how microglia regulate neuronal activity

The C1q-dependent spine pruning by microglia mediates memory forgetting (*Wang et al., 2020*). With increased number of dendritic spines, C1q-deficient mice display enhanced synaptic connectivity and seizure susceptibility (*Chu et al., 2010*). These indicate the importance of C1q in maintaining the neural function. However, we did not observe the alteration of general anesthesia response. Moreover, the spine density was not changed upon the short-term microglial depletion for 14 d. Consequently, our results indicate the microglia-mediated regulation of anesthesia response does not result from dendritic spine. In striatum, microglia serve as a brake suppressing neuronal activity (*Badimon et al., 2020*). P2Y12 knockout in microglia augments the epilepsy susceptible (*Alves et al., 2017*; *Mo et al., 2019*). These evidences indicate that P2Y12 signaling is critical to the stability of neuronal network. However, when we compared neuronal activities by c-Fos staining and patch-clamp recording between the AABRs and EABRs, we observed different consequences to the microglial depletion. With suppressed neuronal activity in AABRs and enhanced neuronal activity in EABRs, microglial depletion results in delayed anesthesia induction and early emergence. Our results also indicate that microglia sophisticatedly and diversely contribute to orchestrating the CNS function, rather than play

an indiscriminate role of negative feedback control as they do in the striatum (*Badimon et al., 2020*). Notably, P2Y12 is downregulated in several neurological disorders (*van Olst et al., 2021*). The consequences of P2Y12 downregulation in neurological disorders are largely unknown. It would be a potential therapeutic target to harness neurological disorders.

## Microglia replacement and general anesthesia

In 2020, we first developed three strategies to achieve efficient microglia replacement (*Xu et al., 2020*), including Mr BMT (*Xu et al., 2020*; *Xu et al., 2021c*), microglia replacement by peripheral blood (Mr PB or mrPB) (*Xu et al., 2020*; *Xu et al., 2021b*), and microglia replacement by microglia transplantation (Mr MT or mrMT) (*Xu et al., 2020*; *Xu et al., 2021a*). We discussed a potential application for treating Alzheimer's disease (AD) by replacing microglia deficient in TREM2, which is one of the major risk factors in sporadic AD (*Jonsson et al., 2013*; *Guerreiro et al., 2013*; *Yuan et al., 2016*; *Leyns et al., 2019*; *Zhong et al., 2019*; *Parhizkar et al., 2019*; *Yeh et al., 2016*; *Wang et al., 2015*; *Lambert et al., 2013*), with TREM2-normal microglia (*Xu et al., 2020*; *Zhang et al., 2023*; *Rao and Peng, 2023*). Recent studies verified this therapeutic effect of Mr BMT in an AD mouse model (*Jiang and Jin, 2023*; *Yoo et al., 2023*), demonstrating the clinical potential of microglia replacement. It also provides new and clinically feasible strategies for treating other neurological disorders (*Xu et al., 2020*; *Zhang et al., 2023*; *Rao and Peng, 2023*; *Shibuya et al., 2022*). Despite sharing similar characteristics with naïve microglia, Mr BMT cells are P2Y12$^-$ (*Xu et al., 2020*). Whether P2Y12$^-$ microglia-like Mr BMT cells influence the response to general anesthesia is unknown. To this end, we tested the response to general anesthesia use in mice treated with Mr BMT and found delayed anesthesia induction and early emergence. This study not only demonstrates the role of P2Y12 signaling in regulating the response to general anesthesia but also identifies an impeded general anesthesia response after Mr BMT treatment.

## Optogenetic and chemogenetic manipulations in neuronal and nonneuronal cells

The 'activation' and 'inhibition' of neurons are defined as the electrical activities (e.g., action potentials) by which neurons convey information and signals. Optogenetics and chemogenetics are powerful tools widely used in manipulating neuronal electrical activity to dissect neural circuitries. Optogenetics relies on light-sensitive ion channels, pumps, or enzymes. Channelrhodopsin-2 (ChR2) is an excitatory optogenetic tool of light-sensitive cation channels from green algae (*Harz and Hegemann, 1991*). Ectopically expressed ChR2 in neurons responds to blue light and undergoes a conformational change, which allows the passive diffusion of Na$^+$, Ca$^{2+}$, H$^+$, and K$^+$. It thus depolarizes the member potential and elicits action potentials in neurons (*Bi et al., 2006*; *Boyden et al., 2005*). Halorhodopsin (NpHR) is an inhibitory optogenetic tool of archaeal light-driven chloride pumps. In response to yellow light, the ectopically expressed NpHR in neurons actively pumps Cl$^-$ into cells and hyperpolarizes the membrane potential (*Zhang et al., 2007*). Chemogenetic tools are based on designer receptors exclusively activated by designer drugs (DREAADs) (*Roth, 2016*). hM3Dq is an excitatory chemogenetic tool of genetically encoded tetracycline-sensitive GPCR (*Alexander et al., 2009*). hM3Dq responds to CNO and activates intracellular G$_{\alpha q}$. Then, the elevated G$_{\alpha q}$ level enhances the intracellular Ca$^{2+}$ concentration, thereby inducing action potentials in neurons. hM4Di is an inhibitory chemogenetic tool (*Zhu et al., 2014*). hM4Di responds to CNO and engages the G$_{\alpha i}$ signaling pathway. G$_{\alpha i}$ in neurons reduces intracellular Ca$^{2+}$ (suppressing presynaptic transmitter release) and opens K$^+$ channels (hyperpolarizing the membrane potential). Thus, the intracellular consequences from optogenetic and chemogenetic manipulations can drive or suppress action potentials in neurons, thus 'activating' or 'inhibiting' the neuron.

In contrast, the nonelectrically excitable cells, including microglia, have no action potentials. The 'activation' (or reactive state) and 'inhibition' are not defined as electrical activities. Instead, 'activation' (or reactive state) and 'inhibition' are defined as responses to specific stimuli in diverse contexts in which nonneuronal cells experience sophisticated alterations. The intracellular events of optogenetic and chemogenetic tools are not directly associated with nonneuronal cell activation. Thus, optogenetic and chemogenetic tools do not simply 'activate' or 'inhibit' nonneuronal cells in the brain. In our study, we ectopically expressed hM3Dq in microglia. Upon CNO administration, hM3Dq elevates intracellular Ca$^{2+}$ levels. It does not result in action potentials in microglia and thus does not 'activate'

microglia. Nonetheless, it is a reliable chemogenetic tool for manipulating the $Ca^{2+}$ level in microglia. We used this approach to investigate the biological function of microglial $Ca^{2+}$ in the response to general anesthesia. Together, optogenetic and chemogenetic tools do not simply 'activate' or 'inhibit' nonneuronal cells. Instead, they can be utilized to study the function of nonneuronal cells regarding specific intracellular events.

## Materials and methods

### Animals

C57BL/6J mice were purchased from SPF (Beijing) Vital River Laboratory Animal Technology. *P2ry12*<sup>fl/fl</sup> mice (*Mo et al., 2019*) were donated by Prof. Jiyun Peng at Nanchang University. *Tmem119*-CreER mice (C57BL/6-Tmem119<sup>em1(cre/ERT2)Gfng</sup>/J, Stock# 031820) (*Kaiser and Feng, 2019*), *Cx3cr1*-CreER mice (B6.129P2(C)-Cx3cr1<sup>tm2.1(cre/ERT2)Jung</sup>/J, Stock# 020940) (*Yona et al., 2013*), *C1qa*<sup>−/−</sup> mice (B6(Cg)-C1qa<sup>tm1d(EUCOMM)Wtsi</sup>/TennJ, Stock# 31675) (*Fonseca et al., 2017*), *β-actin*-GFP mice (C57BL/6-Tg (CAG-EGFP) 131Osb/LeySopJ, Stock# 006567) (*Okabe et al., 1997*), and *LSL*-hM3Dq-YFP mice (B6N;129-Tg(CAG-CHRM3*, -mCitrine)1Ute/J, Stock# 026220) (*Zhu et al., 2016*) were purchased from Jackson Lab. *Stim1*<sup>fl/fl</sup> mice (C57BL/6JGpt-Stim1<sup>em1Cflox</sup>/Gpt, Stock# T013158) were purchased from GemPharmatech. All mice were housed in the Animal Facility at the Department of Laboratory Animal Science at Fudan University under a 12 hr light/dark cycle with food and water given ad libitum. All animal experiments were conducted in accordance with the guidelines of the Institutional Animal Care and Use Committee of the Department of Laboratory Animal Science at Fudan University (202009001S and 2021JS-NZHY-002).

### Chemicals and reagents

PLX5622 was formulated into the AIN-76A diet at a concentration of 1.2 g of PLX5622 per kilogram of diet by SYSE Bio (Cat# D20010801). PLX73086 (Plexxikon) was formulated into the AIN-76A diet at 0.2 g of PLX73086 per kilogram of diet by Research Diet, Inc (Cat# D15180708i). The normal AIN-76A diet (CD) was purchased from SYSE Bio (Cat# PD1001). Chloral hydrate (Cat# C104202) and tamoxifen (Cat# T137974) were purchased from Aladdin. The P2Y12 inhibitor 2-MeSAMP (Cat# HY-125989) and the DREADD agonist CNO (Cat# HY-17366) were purchased from MCE. Xylazine hydrochloride (Cat# X1251) was purchased from Sigma-Aldrich. Propofol (H20123318) was purchased from Xi'an Libang Pharmaceutical. Ketamine (H20193336) was purchased from Shanghai Pharmaceutical. Isoflurane (Lot# 20230501) was purchased from RWD.

### Drug administration

To pharmacologically ablate myeloid cells, mice were administered a PLX5622-formulated AIN-76A diet (1.2 g PLX5622 per kilogram of diet, formulated by SYSE Bio) ad libitum for 14 d. To pharmacologically ablate peripheral macrophages, mice were administered a PLX73086-formulated AIN-76A diet (0.2 g PLX73086 per kilogram of diet, formulated by Research Diet) ad libitum for 14 d. Control mice were fed an AIN-76A CD. Since the microglial ablation efficiency by CSF1R inhibition might be different between sexes (*Shi et al., 2019*), we utilized male mice for this experiment. To efficiently induce CreER-dependent recombination, tamoxifen (150 mg per kg of body weight) dissolved in olive oil (Macklin, 0815210) was administered via oral gavage for four consecutive days following our previously described procedures (*Huang et al., 2018b*; *Zhou et al., 2022*; *Rao et al., 2021*; *Xu et al., 2020*; *Huang et al., 2018a*). 2-MeSAMP (10 mM) was injected into the lateral ventricle 90 min before the behavior test (*Mastorakos et al., 2021*). CNO (100 μg/mL) was administered via intraperitoneal injection 90 min before the behavior test (*Binning et al., 2020*).

### Righting reflex

First, the mice were placed in a box for 5 min for adaptation to the experimental environment. Next, anesthesia was initiated, and the righting reflex of the mice was checked every 15 s from the beginning of anesthesia. When the mice were in an abnormal position (limbs up) and could not voluntarily return to the normal position, this behavior was defined as the LORR. The mouse was kept in a position with its back touching the ground and limbs facing upward during deep anesthesia. A thermostatic heating pad (37°C) was placed under the body to maintain body

temperature. If the mouse automatically returned to the normal position (all limbs touching the ground) from the position where the righting reflex disappeared, it was considered to have recovered. The time from the end of anesthesia to RORR was defined as the time of emergence from anesthesia. All experiments were conducted between 20:00 and 4:00 the next day, in the same light–dark cycle of ZT 12:00 to 20:00.

## Brain tissue preparation

Mice were deeply anesthetized with a mixture of ketamine hydrochloride (100 mg per kg of body weight) and xylazine (10 mg per kg of body weight) by intraperitoneal injection. For histological experiments, animals were sequentially transcranially perfused with 0.01 M PBS and 4% paraformaldehyde (PFA) (Biosharp, Cat# BL539A) in 0.01 M PBS. Brains were then carefully harvested and postfixed in 4% PFA in 0.01 MPBS at 4°C overnight.

## Cryosection preparation

Brains and peripheral organs were dehydrated in 30% sucrose in 0.01 M PBS at 4°C for 3 d. After being embedded in optimal cutting temperature compound (OCT, SAKURA, Cat# 4583), brain and peripheral organ samples were frozen and stored at –80°C before sectioning. Tissue with regions of interest was cut by a cryostat (Leica, CM1950) at a thickness of 35 µm.

## Immunohistochemistry and image acquisition

Brain and peripheral organ sections were rinsed with 0.01 M PBS three times for 10–15 min, followed by blocking with 4% normal donkey serum (NDS, Jackson, Cat# 017-000-121) in 0.01 M PBS containing 0.3% Triton X-100 (Aladdin, Cat# T109026) (PBST) at room temperature (RT) for 2 hr. Then, the samples were incubated with primary antibodies with 1% NDS in PBST at 4°C overnight. After rinsing with PBST for three changes, the samples were incubated with fluorescent dye-conjugated secondary antibodies with 1% NDS in PBST with 4′,6-diamidino-2-phenylindole (DAPI, 1:1000, Sigma-Aldrich, D9542) at RT for 2 hr. Afterward, the samples were rinsed three times before mounting with anti-fade mounting medium (SouthernBiotech, Cat# 0100-01).

Primary antibodies included rabbit anti-IBA1 (1:500, Wako, Cat# 019-19741, Lot# CAJ3125), goat anti-IBA1 (1:500, Abcam, Cat# ab5076, Lot# GR3425808-1), rabbit anti-GFP (1:1000, Invitrogen, Cat# A11122, Lot# 2273763), rabbit anti-c-Fos (1:1000, Abcam, Cat# ab190289, Lot# GR3367372-1), and rabbit anti-P2Y12 (1:500, Sigma-Aldrich, Cat# S5768, Lot# 0000128079). Secondary antibodies included AF488 donkey anti-rabbit (1:1000, Jackson, Cat# 711-545-152, Lot# 161527), AF568 donkey anti-rabbit (1:1000, Invitrogen, A10042, Lot# 2433862), AF568 donkey anti-goat (1:1000, Invitrogen, Cat# A11057, Lot# 2160061), and AF647 donkey anti-goat (1:1000, Jackson, Cat# 705-605-003, Lot# 147708).

Confocal images were acquired by using an Olympus FV3000 confocal microscope with a solid-state laser. Lasers with wavelengths of 405 nm, 488 nm, 561 nm, and 640 nm were used to excite the fluorophores. Then, ×60 (oil), ×40 (oil), and ×20 objectives were utilized. Some whole-brain fluorescence images were acquired by an Olympus VS120 microscope equipped with a motorized stage. Then, ×10 objective was used. Z-stacked focal planes were acquired and maximally projected with Fiji. The brightness and contrast of the image were adjusted with Fiji if necessary.

## c-Fos immunostaining and *Fos* RNAscope dual labeling

Mice were fed a CD or PLX5622 for 14 d and placed alone in a quiet environment for 2 hr, and samples were taken 30 min after intraperitoneal injection of pentobarbital sodium (80 mg/kg body weight). Cryostat sections at 15 µm were collected, and hybridizations were carried out according to the manufacturer's instructions using RNAscope Multiplex Fluorescent Detection Reagents V2 (Advanced Cell Diagnostics, Cat# 323110, Lot# 2015636, 2019446). Briefly, sections were dehydrated in sequential incubations with ethanol, followed by 30 min Protease Plus treatment and RNAscope wash buffer wash. Mouse *Fos* probe (Cat# 316921, Lot# 221048) was incubated for 2 hr at 40°C, followed by three amplification steps. After all these steps, general immunostaining steps were performed as mentioned earlier.

## EEG/EMG surgery and recording

The mice were initially anesthetized by 2% isoflurane and maintained under anesthesia by 1% isoflurane during the surgery. Body temperature was monitored in real time and kept at approximately 37°C throughout the surgical procedure. For the EEG/EMG recording experiment, two stainless steel screws were placed on the prefrontal cortex (recording site) and cerebellar cortex (reference site) as EEG electrodes, and two other thin stainless steel wires were inserted into the bilateral neck muscles as EMG electrodes.

Mice were allowed a minimum of 7 d of recovery following surgery. On the day of recording, the mice were acclimated first for 20 min in a recording box, where the temperature was kept at 25°C, and the mice were allowed to move around freely. All recordings were conducted between 20:00 and 24:00. Signals were amplified (Apollo I, Bio-Signal Technologies, USA) and digitized at a sampling rate of 1000 Hz.

## EEG spectra analysis

The raw EEG signals were downsampled to 250 Hz before analysis. The power spectrum was computed using multitaper methods in the MATLAB Chronux toolbox (version 2.1.2, http://chronux.org/), with 4 s data segments and 3–5 tapers (TW = 3, K = 5). To normalize total power and compare between groups, the power spectra were normalized such that the total area under the spectra was unity (e.g., power spectral density). Power spectral density analysis was performed on the data from the baseline (20 min before injection), induction (slow oscillation appears for the first time and lasts for more than 30 s after injection), and emergence (slow oscillation disappears for the first time and lasts for more than 10 min after deep anesthesia) periods.

The time–frequency power spectrum (by the 'cwt' function in the MATLAB wavelet toolbox) was also computed using 80 Hz downsampled EEG to enhance the temporal resolution.

## Root mean square of EMG

The raw EMGs were further downsampled to 25 Hz. The root mean square (RMS) was obtained using a 20 s moving window.

## Consciousness probability

The vigilance states before and after injection of anesthetic were automatically classified as *awake* and *nonawake* states by using artificial intelligence (AI)-driven software Lunion Stage (https://www.luniondata.com, Shanghai, China) and were checked manually (*Zhai et al., 2023*). The awake probability was generated by 1000 repeat bootstrap analyses, for example, for each repeat, we randomly selected 75% of the total animal data to calculate the percentage of animals in the awake state at each time point. Any epochs considered to contain significant movement artifacts were omitted from the data analysis.

## Intracranial guide tube implantation and microinjection

Briefly, mice were anesthetized with 3% isoflurane (RWD, Lot# 20230501) delivered in 100% $O_2$ and then transferred to a stereotaxic frame with a mouse anesthesia mask (RWD, China). The delivered isoflurane concentration was decreased to 1.5%. A thermostatic heating pad (37°C) was placed under the mouse to maintain body temperature. Unilateral lateral ventricle cannulas were implanted in targeted coordinates (anteroposterior: –0.5 mm; mediolateral: 1 mm; dorsoventral: –2.3 mm) in 8-week-old mice (*Glascock et al., 2011*; *DeVos and Miller, 2013*). After a 7-day recovery from surgery, 5 µL of 2-MeSAMP (10 mM) was injected via the guide cannula using a microsyringe pump at a rate of 0.5 µL/min according to the manufacturer's instructions. Behavioral tests were performed 90 min after the intraventricular injection.

## Microglia replacement by bone marrow transplantation (Mr BMT)

Two approaches were used to achieve myeloid ablation/inhibition of Mr BMT in this study. For Mr BMT by irradiation, 8-week-old recipient mice were fed PLX5622 from day 0 to day 14. Then, the pretreated mice were exposed to 9 Gy X-ray irradiation on day 14 (*Xu et al., 2020*; *Xu et al., 2021c*). For Mr BMT by busulfan, 8-week-old recipient mice were fed PLX5622 from day 0 to day 14. Then, the mice received busulfan (25 mg/kg of body weight for each day) from day 9 to day 12 by intraperitoneal

injection. Afterward, $1 \times 10^7$ BMCs harvested from the tibia and femur of the *β-actin*-GFP donor mouse were immediately introduced into the recipient mice on day 14 via intravenous injection. Then, the mice were fed a CD. The mouse was fed neomycin (1.1 g/L) in acidic water (pH 2–3) throughout the procedure of microglia replacement.

## Acute brain slice preparation for patch-clamp recording and spine quantification

Parasagittal slices containing the LC, SON, and mPFC were obtained from mice aged from postnatal day 60 (P60) to P70. Mice were deeply anesthetized with pentobarbital sodium (80 mg/kg of body weight) before sacrifice by decapitation. The brain was quickly removed and immersed in ice-cold sucrose-based ACSF (10 mM glucose, 213 mM sucrose, 26 mM NaHCO$_3$, 1.25 mM NaH$_2$PO$_4$, 2.5 mM KCl, 2 mM MgSO$_4$, and 2 mM CaCl$_2$). Acute brain slices with a thickness of 300 µm were cut in sucrose-based ACSF by a vibratome (Leica VT 1200S). Afterward, brain slices were immediately transferred to an incubation chamber filled with 95% O$_2$ and 5% CO$_2$ equilibrated normal ACSF (25 mM glucose, 126 mM NaCl, 26 mM NaHCO$_3$, 1.25 mM NaH$_2$PO$_4$, 2.5 mM KCl, 2 mM MgSO$_4$, and 2 mM CaCl$_2$) at 34°C for 45 min. Slices were then transferred to 95% O$_2$ and 5% CO$_2$ equilibrated normal ACSF at RT before recording. Slices were then transferred to a recording chamber continuously perfused in 95% O$_2$ and 5% CO$_2$ equilibrated normal ACSF (approximately 60 mL/hr), with the temperature maintained at 34 ± 1°C. An infrared-differential interference contrast (IR-DIC) microscope (Olympus BX-51WI) was used for visualization of individual neurons.

## Patch-clamp recording

The intracellular solutions contained 138 mM CsCH$_3$SO$_3$, 3 mM CsCl, 2 mM MgCl$_2$, 0.2 mM EGTA, 10 mM HEPES, 2 mM ATP-Na$_2$, and 5 mM QX314. The pH was appropriately adjusted to 7.3 by CsOH, and osmolarity was adjusted to 280–290 mOsm. The electrode impedance was approximately 4–7 MΩ. When recording the evoked EPSCs (eEPSCs), the membrane potential was held at –70 mV. After eEPSC recording, the same cell was held at 0 mV to record evoked IPSCs (eIPSCs). The locations of SON and LC were identified under an IR-DIC microscope based on their location and cell density. The stimulating electrode was placed deep inside the nucleus and approximately 50 µm from the recorded cell. Membrane voltage and current were sampled at 10–25 kHz and low-pass filtered at 2–10 kHz using the patch-clamp amplifier MultiClamp 700B (Molecular Devices, LLC), digitized and sampled by Micro 1401 with Spike2 software (Cambridge Electronic Design) or by Digidata 1440A with pCLAMP 10.2 software (Molecular Devices, LLC). The evoked postsynaptic currents were analyzed using MATLAB R2023a (MathWorks, Inc, Natick, MA) and OriginPro9.1 (Originlab, Inc).

## Biocytin filling and morphological reconstruction

Coronal sections containing the mPFC, SON, and LC were recorded using patch clamp under a whole-cell configuration. The electrode was filled with the patch solution with 0.2% biocytin (Life Technologies, B1592). Neurons that maintained a stable membrane potential for at least 20 min were included. Upon cessation of filling, the pipette was slowly pulled out along the direction of recording until a membrane reseal was formed. After a 10 min recovery, the slices were fixed in 4% PFA overnight at 4°C, cryoprotected in 30% sucrose solution (for 1–3 d), and incubated with AF488 streptavidin (1:1000, Invitrogen, Cat# S11223, 2390711) in PBS containing 0.3% Triton X-100 (Aladdin, T109026) overnight at 4°C. To reconstruct the dendritic spines, the coronal sections were resected at 70 µm thickness and coverslipped with the mounting medium Fluoromount-G (Southern Biotechnology Associates). The images were taken using an Olympus FV3000 confocal microscope equipped with a UPLSAPO ×60 oil-immersion lens (numerical aperture of 1.5). The 2048 × 2048 pixels frame size was used without zooming. Serial Z-stack images with a step size of 0.7 µm were collected. Dendritic segments located 30 µm away from the soma and 50 µm in length were selected for analysis of spine density and category. For individual cells, 8–12 dendritic segments were chosen for analysis. Dendritic length and spines were counted and categorized with ImageJ (NIH). Spines were classified into three subtypes – thin, mushroom, and stubby – based on previously described criteria (*Zhang et al., 2019b*). Briefly, thin spines included a head-to-neck diameter ratio less than 1.1 and a length-to-spine head ratio greater than 2.0. Mushroom spines had a head diameter larger than 0.5 mm and

a head-to-neck diameter ratio greater than 1.1. Stubby spines had no clear border between the head and the attachment to the shaft. Filopodia had a long thin protrusion but without a clear head shape.

### RNA extraction and qPCR

Total RNA from brain tissue was extracted with TRIzol. cDNA was reverse transcribed from total RNA using the Vazyme HIScript III RT SuperMix for qPCR kit according to the manufacturer's instructions. Subsequently, a 20 μL reaction system was prepared for qPCR using Vazyme ChamQ Universal SYBR qPCR Master Mix kit with an ABI StepOne Plus Real-Time PCR system. The relative cDNA concentrations of target genes were normalized to Gapdh. The primers used in this study were synthesized by Tsingke Biotechnology, including:

> *Gapdh*-forward (TGAGGCCGGTGCTGAGTATG),
> *Gapdh*-reverse (TGGTTCACACCCATCACAAACA),
> *Stim1*-forward (CAGGTTCAGTGAGACCCTGTC), and
> *Stim1*-reverse (GCCCACCAAGATCTCCACAA).

### Statistics and reproducibility

The statistical approaches are indicated in the figure legends. For the righting reflex test, block randomization was performed on cages of mice such that an approximate number of mice per cage were assigned to each experimental group. Collection of behavior experiment data was double blinded. c-Fos$^+$/*Fos*$^+$ cell counting and spine morphology analysis were evaluated independently by two blinded experienced researchers with Fiji. No statistical methods were used to predetermine sample sizes, but our sample sizes were similar to those reported in our previous publications (*Huang et al., 2018b*; *Zhou et al., 2022*; *Rao et al., 2021*; *Xu et al., 2020*; *Huang et al., 2018a*; *Niu et al., 2022*). Each experiment was repeated in at least two independent batches to avoid bias among a single batch. The data distribution was assumed to be normal, but this assumption was not formally tested. No data were excluded from the analyses. Data are shown as the mean ± SD or mean ± SEM as specifically identified. The two-tailed unpaired *t*-test, paired *t*-test and one-way or two-way repeated measures (RM) ANOVA followed by Bonferroni's multiple-comparisons test were used to assess statistical significance based on GraphPad Prism 9.0 and MATLAB 2020b (MathWorks, Inc, Natick, MA) if necessary. Significance was defined as $p < 0.05$. The layout of all of the figures was generated using Adobe Illustrato.

### Acknowledgements

The authors thank Lize Xiong (Shanghai Fourth People's Hospital, School of Medicine, Tongji University), Weifeng Yu (Renji Hospital, Shanghai Jiao Tong University), Zhian Hu (Army Medical University), Ji Hu (ShanghaiTech University), Zhe Zhang and Huateng Cao (Institute of Neuroscience, Chinese Academy of Sciences), and for the advice in this study, Jiyun Peng (Nanchang University) for donating the P2ry12$^{fl/fl}$ mice. Bo Peng would like to thank Xiaobai for turning off the laptop when Bo Peng was drafting this and another manuscripts. In addition, the authors express their gratitude and respect to all animals sacrificed in this study. This study was supported by STI2030-Major Projects (2022ZD0204700) (BP), (2022ZD0207200) (YR), and (2021ZD0202500) (YS), National Natural Science Foundation of China (32170958) (BP), (32000678) (YR), (32130044, T2241002) (YS), (32100930) (QH), and (32200953) (WK), 'Shuguang Program' supported by Shanghai Education Development Foundation and Shanghai Municipal Education Commission (22SG07) (BP), Program of Shanghai Academic/Technology Research Leader (21XD1420400) (BP), Shanghai Pilot Program for Basic Research (21TQ014) (BP), The Innovative Research Team of High-Level Local University in Shanghai (BP), Shanghai Municipal Science and Technology Major Project (2018SHZDZX01) (YM), and ZJ Lab (YM).

# Additional information

## Funding

| Funder | Grant reference number | Author |
|---|---|---|
| Ministry of Science and Technology of the People's Republic of China | STI2030-Major Projects 2022ZD0204700 | Bo Peng |
| Ministry of Science and Technology of the People's Republic of China | STI2030-Major Projects 2022ZD0207200 | Yanxia Rao |
| Ministry of Science and Technology of the People's Republic of China | STI2030-Major Projects 2021ZD0202500 | Yousheng Shu |
| National Natural Science Foundation of China | 32170958 | Bo Peng |
| National Natural Science Foundation of China | 32000678 | Yanxia Rao |
| National Natural Science Foundation of China | 32130044 | Yousheng Shu |
| National Natural Science Foundation of China | T2241002 | Yousheng Shu |
| National Natural Science Foundation of China | 32100930 | Quansheng He |
| National Natural Science Foundation of China | 32200953 | Wei Ke |
| Shanghai Municipal People's Government | Program of Shanghai Academic/Technology Research Leader 21XD1420400 | Bo Peng |
| Shanghai Shuguang Program | 22SG07 | Bo Peng |
| Shanghai Municipal People's Government | Shanghai Pilot Program for Basic Research 21TQ014 | Bo Peng |
| Shanghai Municipal People's Government | The Innovative Research Team of High-Level Local University in Shanghai | Bo Peng |
| Shanghai Municipal People's Government | Shanghai Municipal Science and Technology Major Project (2018SHZDZX01) | Ying Mao |

The funders had no role in study design, data collection and interpretation, or the decision to submit the work for publication.

## Author contributions

Yang He, Formal analysis, Investigation, Visualization, Methodology, Writing – original draft, Conceived and designed this study; Taohui Liu, Formal analysis, Investigation, Visualization, Methodology, Writing – original draft; Quansheng He, Software, Formal analysis, Investigation, Methodology, Writing – original draft; Wei Ke, Software, Formal analysis, Investigation, Methodology; Xiaoyu Li, Software, Investigation, Methodology, Writing – original draft; Jinjin Du, Suixin Deng, Zhenfeng Shu, Jialin Wu, Yuqing Wang, Investigation; Baozhi Yang, Formal analysis, Investigation; Ying Mao, Resources, Funding acquisition, Provided necessary study support; Yanxia Rao, Resources, Supervision, Funding acquisition, Project administration, Provided necessary study support; Yousheng Shu, Resources, Data curation, Formal analysis, Funding acquisition, Investigation, Writing – original draft, Writing – review and editing, Conceived and designed this study. Conceptualized this study. Provided necessary study support; Bo Peng, Conceptualization, Resources, Data curation, Formal analysis,

Supervision, Funding acquisition, Validation, Investigation, Visualization, Writing – original draft, Project administration, Writing – review and editing, Accidentally observed an anesthetic-resistant phenotype in microglia-depleted mice in 2015. Conceived and designed this study. Conceptualized this study. Provided necessary study support

### Author ORCIDs
Yang He ⓘ http://orcid.org/0009-0004-9183-818X
Yousheng Shu ⓘ https://orcid.org/0000-0002-2834-2876
Bo Peng ⓘ https://orcid.org/0000-0003-4183-5939

### Ethics
All animal experiments were conducted in accordance with the guidelines of the Institutional Animal Care and Use Committee of the Department of Laboratory Animal Science at Fudan University (permit number: 202110005S).

Reviewer #1 (Public Review): https://doi.org/10.7554/eLife.92252.2.sa1
Reviewer #2 (Public Review): https://doi.org/10.7554/eLife.92252.2.sa2
Reviewer #3 (Public Review): https://doi.org/10.7554/eLife.92252.2.sa3
Author Response https://doi.org/10.7554/eLife.92252.2.sa4

## Additional files

### Supplementary files
• MDAR checklist

### Data availability
All data generated or analyzed in this study, including individual data points, are included in the figures. Source data files for Figures 1–14 have been provided.

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
