## [Editor Report · eLife assessment]

This study presents a **valuable** finding on the mechanisms underlying general anesthesia, with a focus on microglial regulation. The evidence supporting the claims of the authors is **solid**, although some of the novelty of these findings may be reduced based on the recent publication of a similar study. The work will be of interest to medical biologists working on mechanisms of anesthesia, microglia, and neuron–microglia interaction.

---

## [Referee Report · Reviewer #1 (Public Review)]

Summary:

This study by He, Liu, and He et al. investigated the fundamental role of microglia in modulating general anesthesia. While microglia have been previously shown to regulate neuronal network activity, their role in the induction of (i.e., LORR) and emergence from (i.e., RORR) anesthesia has only recently been explored. Recently published work by Cao et al. reported that microglia modulate general anesthesia via P2Y12 receptor. The present study largely reproduces those findings and does so using an impressive array of techniques and clever approaches. Following the serendipitous discovery that microglia-depleted mice exhibit increased LORR and decreased RORR, the authors go on to demonstrate that microglia regulate neuronal activity in a region-specific manner during anesthesia via purinergic receptor-mediated calcium signaling. The manuscript is well written and the data are convincing, elegantly validated using several different methods and controls, and largely complete. Nevertheless, this Reviewer has a few minor comments and suggestions to further strengthen the manuscript.

Strengths:

Impressive number of genetic mouse models, techniques, controls, and methods of validation.

Weaknesses:

Some of the novelty of these findings may be reduced based on the recent publication of a similar study.

---

## [Referee Report · Reviewer #2 (Public Review)]

In this manuscript, He et al. have found that delayed anesthesia induction and early anesthesia emergence were observed in microglia-depleted mice. They also showed that neuronal activities were differentially regulated by microglia depletion, possibly via suppressing the neuronal network of anesthesia-activated brain regions and activating emergence-activated brain regions. Mechanistically, this influence was found to be dependent on the activation of microglial P2Y12 receptors and subsequent calcium influx. These findings contribute to a better understanding of the role microglia play in regulating anesthesia and shed light on the underlying mechanisms involved. Nonetheless, there are still some aspects that require further investigation and clarification.

1. In Figure 3A the authors used IBA1 to represent microglia, and the corresponding description is 'brain microglia were not influenced'. However, IBA1 is not a specific biomarker for brain resident microglia. It's recommended to use other biomarkers, such as TMEM119 and P2RY12 to better examine the efficiency of microglial depletion.

2. In Figure 7, 8 and 9 the authors stated that they aim to investigate the impacts microglia exert on neuronal activity. However, using only c-Fos is not sufficient to represent neuron. The authors are supposed to combine c-Fos with other specific biomarkers for neuron to better validate their conclusions.

3. In Figure 11 the authors use C1qa-/- transgenic mice and draw the conclusion 'microglia mediated anesthesia modulation does not result from spine pruning'. However, as C1q contains multiple subtypes, I have some reservations regarding whether the authors' conclusion is entirely warranted based solely on the knockout of a single subtype of C1q.

4. In Figure 14E the authors showed that expression levels of Stim1 is significantly down-regulated in CX3CR1CreER::STIM1fl/fl mouse brains. While this is not incorrect, I would suggest the authors sort microglia with FACS or MACS to perform q-RT-PCR and examine the expression levels of Stim1 since the Cre-LoxP system here is microglia specific.

5. The flow of the manuscript should have been improved. For instance, the results of repopulated microglia in Figure 1B was described even after Figure 2 and 3, which makes the manuscript a little confusing. Additionally, in Figure 14, it would be beneficial to provide a more comprehensive introduction to molecules such as hM3Dq and Stim1 to improve the clarity and readability of the result descriptions.

---

## [Referee Report · Reviewer #3 (Public Review)]

Summary:

This work aims to understand the contribution of microglia to anesthesia induced by general anesthetics. The authors report that ablation of microglia shortens anesthesia, manifested by the delay of anesthesia induction and the early anesthesia emergence. They show that microglial depletion suppresses activity in the neuronal network of anesthesia-activated brain regions but enhances activity in emergence-activated brain regions. Based on these findings, the authors suggest microglia facilitate and stabilize the anesthesia status. To elucidate the underlying mechanism, they further tested the potential contribution of microglia-mediated dendritic spine plasticity and microglial P2Y12-Ca2+ signaling, and identified the latter as a critical pathway through which microglia regulate anesthesia.

Strengths:

A major strength of this study is the systematic experimental design, which includes multiple anesthetics and complementary approaches, leading to very compelling data. As a result, a significant contribution of microglia in instating and maintaining the state of anesthesia is convincingly established. In addition, the results also shed light on the potential underlying microglial mechanistic. The findings are of relevance to both medical practice and basic understanding of microglial biology and neuron-glia interactions.

Weaknesses:

The study produces a large amount of data that is in general cohesive and support the main conclusions, but more thorough considerations on some of their findings may be helpful, as exemplified by the following:

1. the effect of microglial ablation on chloral hydrate-induced RORR in Fig. 1B appears to be not the same as other anesthetics. what does this mean?

2. Macrophage ablation impedes anesthesia emergence from pentobarbital (Fig. 3C). how may this occur?

3. examination of the potential effect of microglial depletion on dendritic spine density is interesting but the experimental design does not seem to align well with the PPR and eEPSC data, which indicate a reduction in presynaptic release (Fig.10E) and increase of postsynaptic function (Fig. 10H), respectively. The PPR data seems to suggest a presynaptic effect of microglia; ablation.

---

## [Author Response]

**Reviewer #3 (Public Review):**
Summary:This work aims to understand the contribution of microglia to anesthesia induced by general anesthetics. The authors report that ablation of microglia shortens anesthesia, manifested by the delay of anesthesia induction and the early anesthesia emergence. They show that microglial depletion suppresses activity in the neuronal network of anesthesia-activated brain regions but enhances activity in emergence-activated brain regions. Based on these findings, the authors suggest microglia facilitate and stabilize the anesthesia status. To elucidate the underlying mechanism, they further tested the potential contribution of microglia-mediated dendritic spine plasticity and microglial P2Y12-Ca2+ signaling, and identified the latter as a critical pathway through which microglia regulate anesthesia.Strengths:A major strength of this study is the systematic experimental design, which includes multiple anesthetics and complementary approaches, leading to very compelling data. As a result, a significant contribution of microglia in instating and maintaining the state of anesthesia is convincingly established. In addition, the results also shed light on the potential underlying microglial mechanistic. The findings are of relevance to both medical practice and basic understanding of microglial biology and neuron-glia interactions.Weaknesses:The study produces a large amount of data that is in general cohesive and support the main conclusions, but more thorough considerations on some of their findings may be helpful, as exemplified by the following:1. the effect of microglial ablation on chloral hydrate-induced RORR in Fig. 1B appears to be not the same as other anesthetics. what does this mean?2. Macrophage ablation impedes anesthesia emergence from pentobarbital (Fig. 3C). how may this occur?3. examination of the potential effect of microglial depletion on dendritic spine density is interesting but the experimental design does not seem to align well with the PPR and eEPSC data, which indicate a reduction in presynaptic release (Fig.10E) and increase of postsynaptic function (Fig. 10H), respectively. The PPR data seems to suggest a presynaptic effect of microglia; ablation.

This reviewer may confused the brain regions between our spine quantification (Figure 11) and patch-clamp recording (Figure 10). In our spine quantification, all evaluations were conducted in the mPFC. However, the patch-clamp recording were performed in SON (Figure 10 B-F) and LC (Figure 10 G-K), different brain regions from our spine quantification. As one of our conclusion, microglia differentially modulate the activity of neuronal network in a brain region-specific manner, neurons in different brain regions may exhibit different electrophysiological alterations upon microglial depletion. Therefore, this comment might be a factual error.